# Anthropogenically-driven increases in the risks of summertime compound hot extremes

Jun Wang [1], Yang Chen [2]*, Simon F.B. Tett [3], Zhongwei Yan[1,4], Panmao Zhai[2], Jinming Feng[1] & Jiangjiang Xia[1]

Compared to individual hot days/nights, compound hot extremes that combine daytime and nighttime heat are more impactful. However, past and future changes in compound hot extremes as well as their underlying drivers and societal impacts remain poorly understood. Here we show that during 1960–2012, significant increases in Northern Hemisphere average frequency (~1.03 days decade$^{-1}$) and intensity (~0.28 °C decade$^{-1}$) of summertime compound hot extremes arise primarily from summer-mean warming. The forcing of rising greenhouse gases (GHGs) is robustly detected and largely accounts for observed trends. Observationally-constrained projections suggest an approximate eightfold increase in hemispheric-average frequency and a threefold growth in intensity of summertime compound hot extremes by 2100 (relative to 2012), given uncurbed GHG emissions. Accordingly, end-of-century population exposure to compound hot extremes is projected to be four to eight times the 2010s level, dependent on demographic and climate scenarios.

[1] Key Laboratory of Regional Climate-Environment for Temperate East Asia (RCE-TEA), Institute of Atmospheric Physics, Chinese Academy of Sciences, Beijing 100029, China. [2] State Key Laboratory of Severe Weather, Chinese Academy of Meteorological Sciences, Beijing 100081, China. [3] School of GeoSciences, The University of Edinburgh, Edinburgh EH9 3FF, UK. [4] University of Chinese Academy of Sciences, Beijing 100049, China. *email: ychen@cma.gov.cn

It is well known that hot extremes, during the hottest season in particular, have adverse societal and environmental impacts[1–4]. In a warming climate, increasingly frequent and intense hot extremes have been reported globally with strong evidence pointing to a large contribution from anthropogenic warming[5–8]. Severe damage comes from sequential occurrences of hot day and hot night within 24 h, which accumulate and aggravate adverse impacts of daytime and nighttime heat on various sectors[9,10]. Some studies considered both diurnal and nocturnal temperatures, for instance using daily mean temperature as a measurement[11,12]. However, compared with the well-understood univariate hot days and hot nights[7,8,13,14], current knowledge about combined daytime–nighttime hot extremes remains too sparse to inform development of type-specific adaptation and mitigation strategies.

Combined daytime–nighttime hot extremes might differ from individual hot days/nights not only in meteorological and climatological aspects[15–17] but more importantly in impacts on human and natural systems[18]. Specifically, combined events are reportedly more damaging to human health, as the ensuing nighttime heat deprives humans of their chance to recover from the preceding daytime heat[19,20]. Overlooking this compounding effect may lead to serious underestimate of heat-induced consequences. Hence, it is worthwhile to revisit observation, detection–attribution and projection of hot extremes based on a bivariate definitional framework, to refine and further advance our understandings about their past changes and underlying drivers as well as future impacts and risks[21].

To this end, we first define three nonoverlapping types of summertime hot extremes, i.e., independent hot days (daytime events, hot day–mild night), independent hot nights (nighttime events, mild day–hot night), and compound hot extremes (hot day–hot night, see the Methods section). With respect to these bivariate-classified hot extremes, we conduct a series of analyses on their historical changes, mechanism explanations, quantitative detection and attribution, constrained projections, and future population exposure. We find that across Northern Hemisphere lands, the rise in anthropogenic greenhouse gases has driven summertime compound hot extremes increasingly frequent and intense from 1960 to 2012, with those trend patterns closely linked to regional nocturnal land–atmosphere coupling strengths. At the end of the 21st century, uncurbed greenhouse gases emissions would make three-quarters of summer days typical of today's compound hot extremes, leading to several-fold growth in population exposure to them.

## Results

### Observed changes in compound hot extremes.
Summertime compound hot extremes' frequency and intensity (see the Methods section) have exhibited significant increases across most of the mid–high latitudes during 1960–2012 (Fig. 1). Larger increases in frequency are observed in southern parts of the United States, Northwest and Southeast Canada, Western and Southern Europe, Mongolia, and Southeast China, while stronger intensifications occur in the Southwest United States, Northern and Southeast Canada, and broad swaths of Eurasia. The HadGHCND[22]-based spatial–temporal trend patterns are consistent with those based on the Berkeley Earth Surface Temperature data set[23] (Supplementary Fig. 1). This indicates the robustness of trend estimates against the choice of data sets that differ markedly in homogenization levels, data sources, and pre-processings. The robustness of trend estimates is also underpinned by their insensitiveness to the choice of analysis periods (Supplementary Fig. 2).

By contrast, trends for independent hot days are weaker, less significant, and more spatially heterogeneous (Fig. 1c, d). Thus, previous estimates of traditionally defined hot days' trends, which reflect a mixture of changes in compound events and independent hot days, actually underrepresent (overrepresent) the greater (smaller) rate (% decade$^{-1}$) and higher (lower) significance of frequency/intensity increases in compound hot extremes (independent hot days) (Supplementary Fig. 3a–d). Independent hot nights have also experienced significant increases in frequency and intensity across the Northern continents, but with a smaller intensification rate compared with compound hot extremes (Supplementary Fig. 3).

Observed trend patterns for the frequency of hot extremes are basically captured by the multi-model ensemble (MME) mean, as evidenced by significant pattern correlations between them (Supplementary Fig. 4). The reductions in independent hot days in southern Canada and central–eastern China, however, fail to be reproduced, possibly due to models' misrepresentation of key local-scale processes cooling Tmax there (e.g., expansion of irrigation and crop planting in both regions[24,25], and increasing aerosols in central–eastern China[26]). The simulated trends' inaccuracy, particularly in intensity at local-to-regional scales, may also be linked to considerable smoothing of internal variability by the multi-model ensemble mean[27,28].

### Statistical and physical mechanisms.
Before formal detection and attribution, we explore respective roles of summer-mean temperature rise (i.e., general warming) and changing temperature variability in determining changes in summertime compound hot extremes. We do this by re-computing frequency and intensity trends after removing the general warming signal (see the Methods section). We find that the summer-mean warming over 1960–2012 largely dictates the past increases in frequency and intensity of compound hot extremes during that period in both observations and simulations (Fig. 2). By dissecting the contribution from each parameter (e.g., location mean, scale variability, and shape width of tail) of daily temperature distributions (Supplementary Note 1, Supplementary Fig. 5), we confirm that the increase in frequency of compound hot extremes results primarily from the general warming of boreal summer as expressed by a positive shift of the location parameter.

Observed trends for compound hot extremes show marked regional differences and greater magnitudes compared to other types in some areas (Fig. 1; Supplementary Fig. 3). To explain this geographical heterogeneity, we examine the dependence of compound hot extremes' changes on regional physical processes (Fig. 3). Theoretically, anticyclonic setups facilitate greater adiabatic heating and more absorbed solar radiation. These conditions bring higher Tmax and also store more heat near the surface, thus partly offsetting the nighttime radiative cooling and elevating Tmin[17]. An increase in anticyclonic conditions should lead to an increase in compound hot extremes. We calculate trends for both sea-level pressures and 500 hPa geopotential heights to approximate unforced and warming-forced circulation changes[29]. Increasing occurrences of anticyclonic conditions are found especially pronounced in Europe, southeastern Greenland, western Asia, and northeastern Asia (Supplementary Fig. 6, see synoptic-scale analysis in refs. [30,31]). So, regions observing stronger increases in anticyclonic conditions generally see larger increases in frequency of compound hot extremes (compare Supplementary Fig. 6a, b with Fig. 1a), with this relationship more significant using 500 hPa geopotential height trends (Fig. 3b, c). After accounting for strong influences of the general warming on 500 hPa geopotential height increases and the possible bias in reanalysis data, however, the evidence that

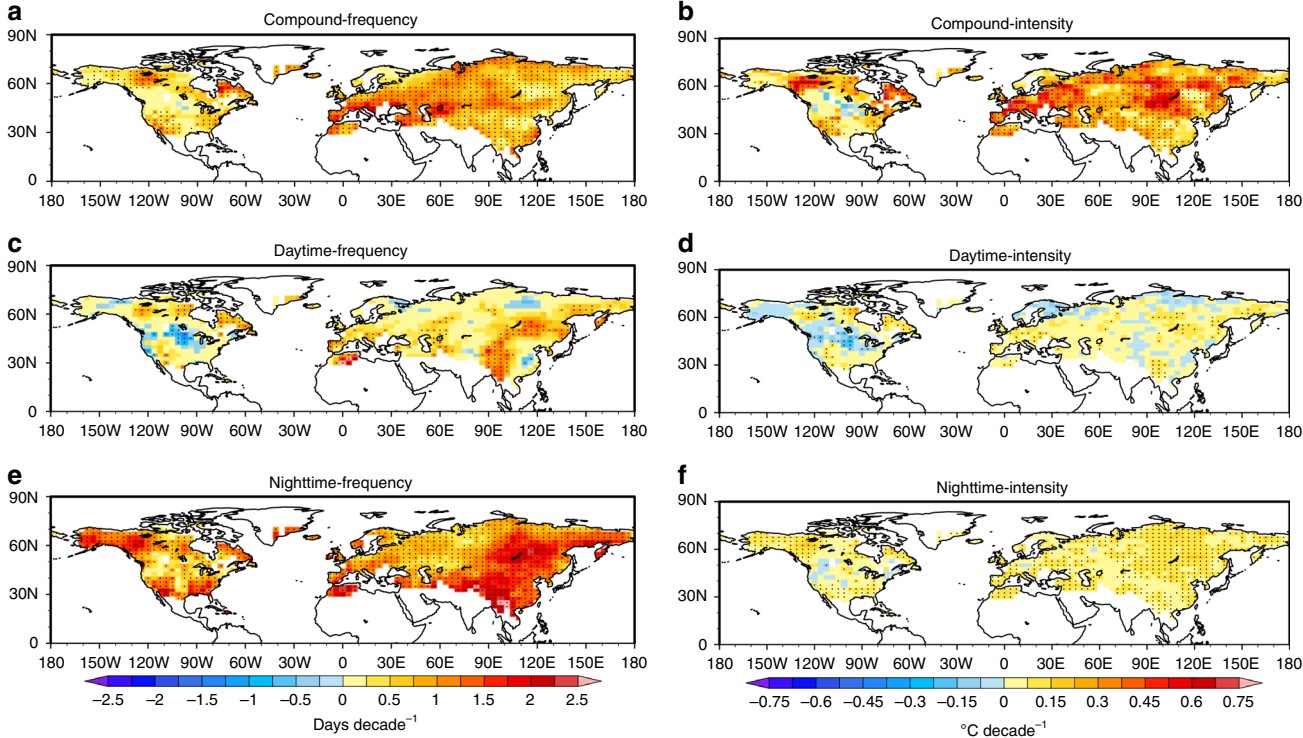

**Fig. 1 Observed changes in summertime hot extremes.** Linear trends for frequency and intensity are estimated for the period of 1960–2012 based on the HadGHCND observations, with respect to compound hot extremes (**a**, **b**), independent hot days (**c**, **d**), and independent hot nights (**e**, **f**). Stipples indicate significance at the 0.05 level.

increases in compound hot extremes have been dynamically contributed by increasing presence of anticyclonic conditions seems not as strong as theoretically expected (Fig. 3c).

Drying soil has also been proposed as an important driver for not only daytime hot extremes[32,33] but also extreme hot conditions at night[34,35], implying that regions of stronger land–air interactions may see larger increases in compound hot extremes. We use the correlation between detrended precipitation and detrended temperatures (Tmax & Tmin) to measure the strength of soil moisture–air temperature coupling[36,37]. Negative correlations occur where enhanced sensible heat fluxes from drier soil bring higher air temperature. Increases in compound hot extremes are larger in areas with stronger nocturnal land–air interactions (compare Supplementary Fig. 6c with Fig. 1a), and such a physical linkage is statistically significant (Fig. 3d). By contrast, despite a more uniform pattern of anticorrelation between Tmax & precipitation (Supplementary Fig. 6d), stronger daytime land–air interaction alone does not necessarily induce greater increases in compound hot extremes (Fig. 3e). Stronger nocturnal land–air interactions are co-located with greater increases in anticyclonic activities in some hotspots for frequency increases (Fig. 3b–d, red and green symbols). This implies the joint role of these two physical processes in strengthening the coupling between daytime and nighttime hot extremes (Supplementary Fig. 7), partly explaining greater increases in compound events than decoupled hot days/nights there.

Considering the well-established causal linkage between the general warming and anthropogenic emissions of GHGs[5], we may qualitatively infer an important role of human-induced global warming in these observed changes. This is also underpinned by the similarity between the observed trend pattern driven by the general warming (Fig. 2a, b) and the forced pattern as simulated by the multi-model ensemble mean (Supplementary Fig. 4a, b). Even so, formal detection and attribution analyses are still needed

to quantitatively evaluate contributions of different external forcings (e.g., GHGs, anthropogenic and volcanic aerosols), which help to pin down the main driver for past changes in compound hot extremes[38–40] and allow calibration of future projections (see the Projection section below). Quantitative attributions and reliable projections are desired by policy-makers to devise strategies to alleviate future impacts and risks from compound hot extremes.

**Detection and attribution.** The hemispheric-average frequency and intensity of summertime compound hot extremes have significantly increased by 1.03 days decade$^{-1}$ (90% confidence interval (CI): 0.82–1.26 days decade$^{-1}$) and 0.28 °C decade$^{-1}$ (90% CI: 0.23–0.33 °C decade$^{-1}$) during 1960–2012 (Fig. 4). These increases are qualitatively well reproduced by simulations with all forcings included.

We use an optimal fingerprinting approach[38] (see the Methods section) to estimate contributions from anthropogenic (ANT) and natural forcings (NAT) to the observed hemispheric-scale changes in summertime compound hot extremes. As shown in Fig. 5a, the significant departure of scaling factors for ANT and NAT from zero signifies the detection of these external forcings. For both frequency and intensity changes, a best-estimated scaling factor slightly larger than one is required to amplify simulated responses to ANT forcings to best match observations (Fig. 5a). A three-signal analysis supports this detection statement and further highlights the dominance of anthropogenic emissions of GHGs in the detectability of ANT forcings. By contrast, a failure to detect other anthropogenic forcings (OANT, dominated by anthropogenic aerosols and large-scale land-use changes[6]) is indicated by the inclusion of zero within the uncertainty range of their scaling factors.

Quantitatively speaking, the human-induced rise in GHG concentration contributes the most to the past increases in

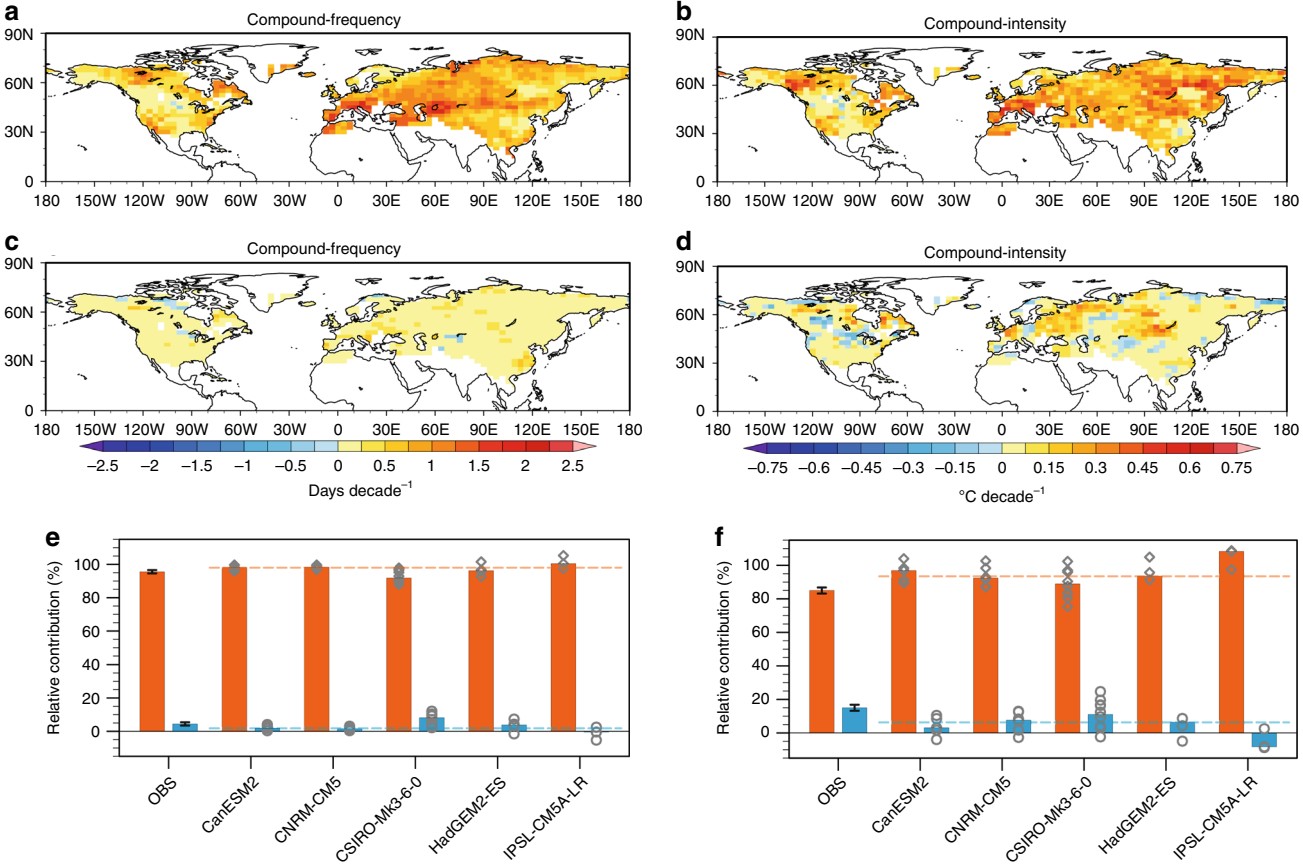

**Fig. 2 Contributions from changing temperature mean and variability.** Observed changes in frequency and intensity of compound hot extremes caused by changes in summer-mean temperature are shown in **a**, **b** and those caused by changes in temperature variability are displayed in **c**, **d**. **e**, **f** show observed and modeled ensemble median contributions from changing summer-mean temperature (orange bars) and temperature variability (blue bars) to area-weighted mean frequency (**e**) and intensity (**f**) changes, respectively. The vertical black bars show the 5–95% uncertainty range of contributions in observations. Gray diamonds and circles indicate values from individual simulations of each model, with their MME (multi-model ensemble) median shown by orange and blue dashed lines.

compound hot extremes, in the frequency of 1.18 days decade$^{-1}$ (5–95% uncertainty range (UR): 0.96–1.41 days decade$^{-1}$) and in the intensity of 0.28 °C decade$^{-1}$ (5–95% UR: 0.22–0.34 °C decade$^{-1}$) during 1960–2012 (Fig. 5c). These GHG-forced increases are a little offset by the cooling effect of OANT forcings, with a best estimate of −0.09 days decade$^{-1}$ (5–95% UR: −0.20–0.03 days decade$^{-1}$) for the frequency and −0.02 °C decade$^{-1}$ (5–95% UR: −0.04–0.01 °C decade$^{-1}$) for the intensity. Thus, anthropogenic emissions of GHGs should have produced around 7–8% larger increases in frequency and intensity of compound hot extremes than observed. Despite the detection of NAT's role (Fig. 5a, b), the attributable portion from it to both frequency and intensity increases is far less than that from anthropogenic GHGs (Fig. 5c). These detection and attribution conclusions are robust against alternative time-smoothing schemes, such as using 5-year-mean instead (see the Methods section and Supplementary Fig. 8).

The same methodology is also applied to detect and attribute observed changes in independent hot days and nights (see Supplementary Note 3). Both ANT and NAT signals are detected in observed changes of these two types of summertime hot extremes (Supplementary Figs. 9, 10). The historical simulations overestimate (underestimate) responses of independent hot days (nights) to anthropogenic GHGs, thus warranting a scaling factor below (above) the unity to scale down (up) simulated responsive changes.

**Observationally constrained projections.** Aforementioned varying degrees of underestimations/overestimations of modeled responses to external forcings would bias projections of hot extremes, if simply extrapolating un-scaled responses to pre-scribed emission levels in the future (e.g., RCP4.5 and RCP8.5). We take advantage of observation-based calibration on responses to external forcings to constrain projections (ref. 40, also see the Methods section). Compound hot extremes show the greatest increases in frequency and intensity (Fig. 6); while the frequency is projected to stay nearly constant for independent hot days, and to increase gradually under RCP4.5 and to peak then fall under RCP8.5 for independent hot nights. These distinct increases in hot extremes' frequency result in drastic shifts of the most common type of summertime hot extremes, an impact-relevant character underreported previously. Specifically, the dominance of independent hot days in total hot extremes before the 1990s has been replaced by independent hot nights, whose dominance is expected to hold till the 2030s (Fig. 6a, c). After that, compound hot extremes become the most common type across the Northern continents. This rapid transition calls for urgent adaptation and mitigation efforts against compound hot extremes in particular. Relative to 2012, anthropogenic forcings will cause an approximate fourfold increase in the hemispheric-average frequency of compound hot extremes (from 8.3 days per summer to 32.0 days per summer) under RCP4.5 by the end of the 21st century. Following a high-end emission pathway (RCP8.5), about

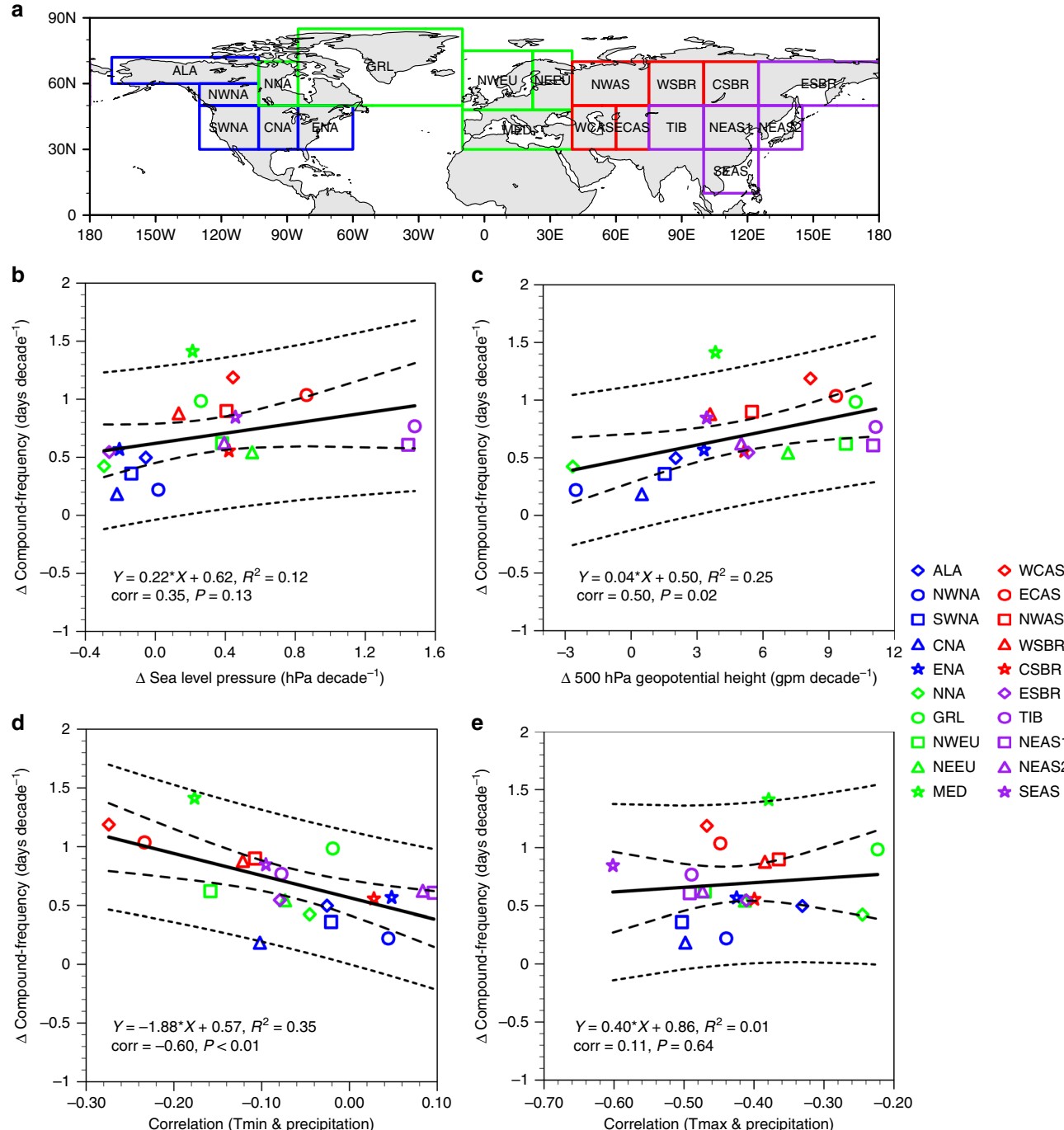

**Fig. 3 Dependence of trend patterns on physical drivers. a** Climate zones and their acronyms. **b**, **c** Scatterplot between trends for circulation changes represented by (**b**) sea-level pressure and (**c**) 500 hPa geopotential height and frequency trends for compound hot extremes averaged in each of the twenty climate zones during 1960–2012. **d**, **e** Scatterplot between summertime monthly-mean daily minimum (**d**) and maximum (**e**) temperature–precipitation correlation and frequency trends for compound hot extremes during 1960–2012. Before calculating correlation coefficients, both monthly-mean temperature and precipitation series are linearly detrended. Each symbol represents one climate zone. Long and short dashed lines show the 95% confidence and prediction intervals for the regression, respectively. The linear regression equation, the proportion of the variance of Y explained by X ($R^2$), the Pearson correlation coefficient (corr), and its p-value (P) are indicated in each panel. For calculation details for **b** and **c** see Supplementary Note 2.

three-quarters of summer days (~69 days) would be compound hot extremes before 2100, equivalent to over an eightfold increase.

Converting these emission pathways to specific warming levels (Methods), we find that compared with a 1.5 °C warmer world, 2 °C of global warming signifies, on average across the Northern Hemisphere lands, an extra ~5 days of compound hot extremes and an additional ~0.5 °C increase in their intensity. However,

4–6 °C of global warming from the non-mitigated pathway (RCP8.5) adds extra 40–60 days in frequency and 4–6 °C in intensity of compound hot extremes, relative to the 1.5 °C status (Fig. 6c, d). Of note, the hemispheric-average intensity of compound events increases quasi-linearly with the rising levels of global warming in the future, indicative of a decisive role of general warming[41]. This consolidates and extends

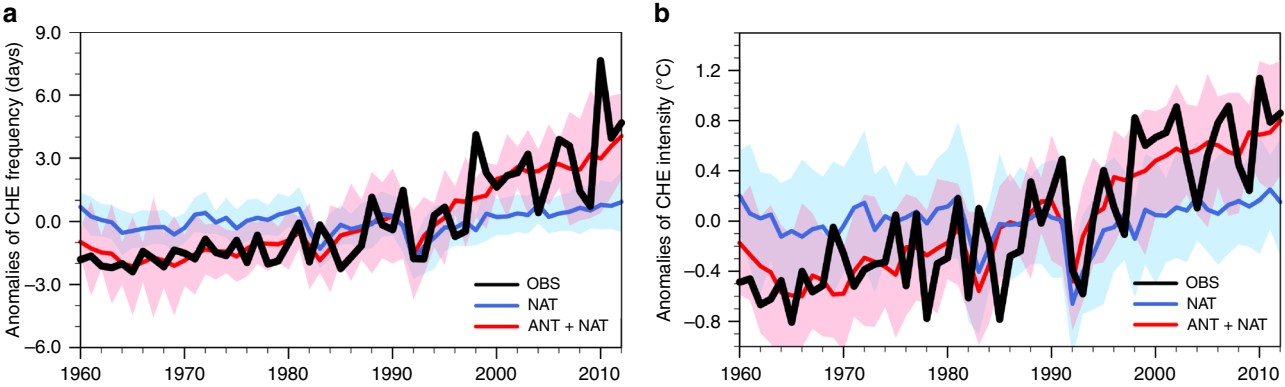

**Fig. 4 Hemispheric-average indices of compound hot extremes over 1960–2012. a** Anomalies in area-weighted mean frequency. **b** Anomalies in area-weighted mean intensity. All anomalies are relative to the 1960–2012 mean. Shown include observations (black line); the MME (multi-model ensemble) mean simulations forced jointly by ANT (anthropogenic) and NAT (natural) forcings (ALL; red line) and the 5–95% range of ALL responses among individual simulations (red shading); and the MME mean simulations forced only by NAT forcings (blue line) with the 5–95% range of NAT responses among individual simulations (blue shading).

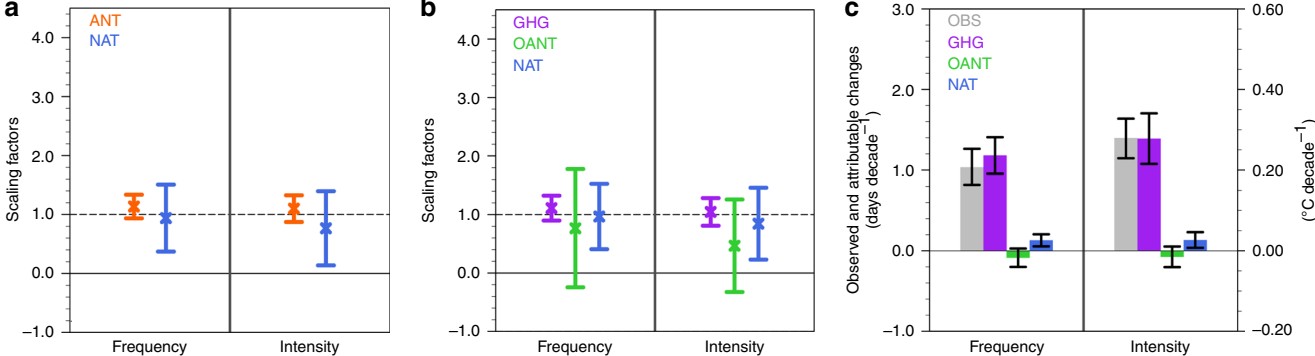

**Fig. 5 Scaling factors and attributable changes for compound hot extremes. a** The best estimate (cross) and 5–95% uncertainty range (bar) of scaling factors for ANT (anthropogenic, orange) and NAT (natural, blue) forcings. **b** Same as **a** but for GHG (greenhouse gases, purple), OANT (other anthropogenic, green), and NAT (blue) in the three-signal detection analysis. **c** The best estimate (shading) for observed changes (gray) and those changes attributable to GHG (purple), OANT (green) and NAT (blue), with black bars representing the 90% confidence interval for observed trends and the 5–95% uncertainty range for attributable trends. The calculations of confidence interval for observed trends and the uncertainty range for attributable changes are detailed in the Methods section. For the meaning of scaling factors and attributable changes see the Methods—Formal detection and attribution section.

observation-based estimates (Fig. 2f). Also notable is that the compound type is the only one showing monotonic increases in frequency and intensity with rising levels of GHGs and global mean surface air temperature (GMST).

Subject to scaling factors' calibration, the range of simulated historical changes now better encapsulates observed counterparts and the MME mean is much closer to the observations (compare Supplementary Fig. 11 with Supplementary Fig. 12). This improvement of consistency between simulations and observations is particularly pronounced in compound and nighttime events. For both types, the divergence between uncalibrated and calibrated projections augments with higher levels of GHG emissions and GMST. Under RCP8.5, by the end of the 21st century, constrained MME mean projection of compound event frequency (intensity) is ~13% (~8%) larger than the default MME mean. The combination of bivariate classification and constrained projection, therefore, warns about higher risks of summertime compound hot extremes than originally predicted.

**Future population exposure to compound hot extremes.** We assess future population exposure[42] (Methods) to heat hazards by combining climate projections and population projections

compatible with Shared Socioeconomic Pathways (SSPs)[43]. Even if the world evolves toward a sustainable future via moderately mitigated GHG emissions (RCP4.5) and low population growth (SSP1), the Northern Hemisphere still expects to see nearly a quadrupling of population exposure to compound hot extremes, from 19.5 billion person-days in the 2010s to 74.0 billion person-days in the 2090s (Fig. 7a). By contrast, the scenario combining unmitigated emissions (RCP8.5) and rapidly growing populations (SSP3) is projected to see an over eightfold increase to 172.2 billion person-days in the 2090s (Fig. 7b). Greater increases are clustered over highly urbanized and/or populous regions, such as eastern United States, western Europe, western Asia, and eastern China (Supplementary Fig. 13). Population exposure to daytime and nighttime hot extremes exhibits a similar peak structure, with the differential exposure to them in two worlds (RCP4.5 and SSP1 vs. RCP8.5 and SSP3) substantially smaller than that to compound type (Fig. 7; Supplementary Fig. 13). After 2030, the compound type would be the one that populations in the Northern Hemisphere are most frequently exposed to (Fig. 7).

The high similarity in temporal patterns of hazard (Fig. 6) and exposure (Fig. 7) demonstrates the dominant role of anthropogenically driven increases in hot extremes in determining increases in the hemispheric-scale population exposure. However,

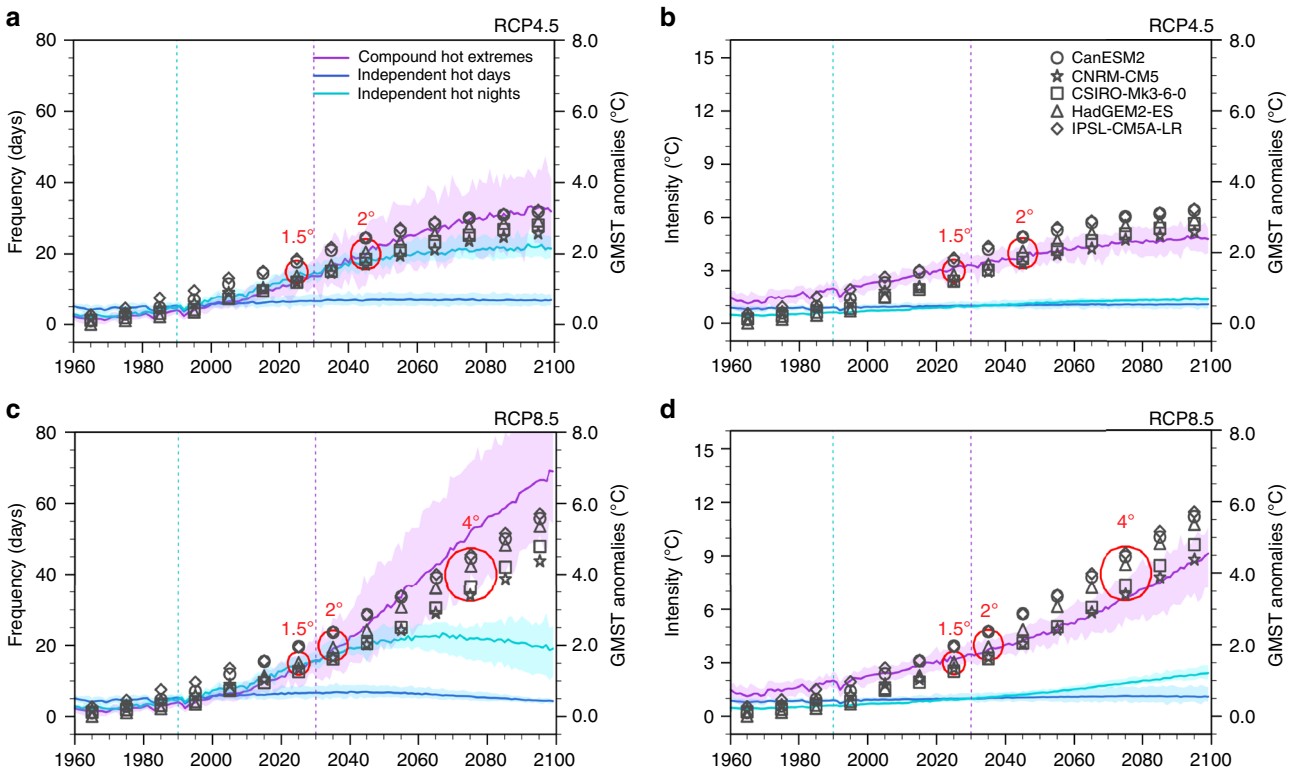

**Fig. 6 Constrained projections of summertime hot extremes.** Area-weighted series of simulated and projected MME (multi-model ensemble) mean frequency (**a**) and intensity (**b**) of summertime compound hot extremes (purple lines), independent hot days (blue lines), and independent hot nights (green lines) under RCP4.5. **c**, **d** Same as **a**, **b**, but under RCP8.5. Shadings enclose the 5–95% range of individual simulations for each type. Black symbols represent decadal-average GMST (global mean surface air temperature) anomalies (relative to 1861–1890, right y-axis) from five used models, with their names specified by the legend in **b**. Red circles enclose the MME mean of decadal-average GMST anomalies, the average among which reaches specific levels of global warming at 1.5 °C, 2 °C, and 4 °C. Two vertical dashed lines locate the years of 1990 and 2030, when the transitions of the dominant type of summertime hot extremes occur.

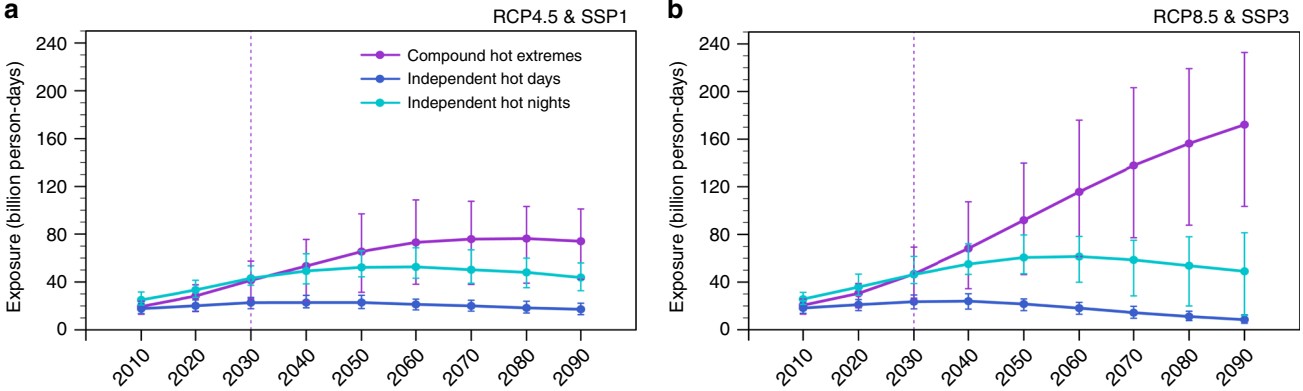

**Fig. 7 Projections of population exposure to summertime hot extremes. a** Population exposure to summertime compound hot extremes (purple lines), independent hot days (blue lines), and independent hot nights (green lines) across the Northern continents through the 21st century in the integrated scenario combining RCP4.5 (climate) and SSP1 (population) for a future with relatively low adaptation and mitigation challenges. **b** Same as **a**, but in the integrated scenario constituted by RCP8.5 (climate) and SSP3 (population) for a future with rapid growth in both greenhouse gas emissions and populations. Decadal-average MME (multi-model ensemble) means are indicated by dots connected by solid curves, with vertical bars framing the 5–95% range of all members' projections. The vertical dashed line locates the year of 2030, after which compound hot extremes will become the type that populations in the Northern Hemisphere are most frequently exposed to.

above estimates in population exposure only present a lower boundary, since the raw climate projections that we use for calculating exposure (rationale see the Methods section) underestimate future increases in compound heat hazards as addressed above. Underestimation in population exposure to compound hot extremes also arises from the insufficient land coverage in the

analysis, with some highly populous areas like India unaccounted for (Supplementary Fig. 13).

## Discussion

In this study, we report observed changes in compound hot extremes across the Northern continents, with underlying

mechanisms proposed and contributions from various external forcers quantified. On this basis, future changes in both heat hazards and population exposure to them are projected. These findings provide new insights into heat-related risk assessment and management. Added value in guiding adaptation and mitigation planning could be gained by further considering the vulnerability of various communities and sectors to these hot extremes. This better embracement of the risk framework calls for a closer multidisciplinary collaboration by sharing the data, methodology and knowledge among different fields. It is reasonable to expect that compound hot extremes are more dangerous to human health[12], agriculture[44], and ecology fields[45], as this type impairs human and natural systems' resilience to ambient excess heat.

The limited data availability over much of the Southern Hemisphere prohibits us from conducting a quasi-global-scale analysis. Although the Berkeley Earth Surface Temperature data set[23] provides a global coverage by merging 14 databases of station observations, the data quality and availability still vary apparently with time and region, particularly at a daily scale critical to identify extremes. We also stress that the quality of observational data matters for detection–attribution–projection conclusions, even though the homogenized Berkeley data[23] and non-homogenized HadGHCND[22] provide very similar area-weighted time series at a hemispheric dimension here. Influences of data quality on detection–attribution–projection, however, may stand out more starkly in regional-scale analysis (e.g., Supplementary Fig. 1e, f).

Although previous studies have highlighted the importance of increasing summer-mean temperatures to hot day or night changes[46,47], this is the first study confirming the dominant role of general warming in observed increases in compound hot extremes. There are contrasting evidences indicating that changes in temperature variability also played an important or even determinant role in inducing changes in hot extremes at regional scales (e.g., North America)[48,49] or in producing extraordinarily intense cases[50]. These inconsistencies may stem from different data sets and methods used to quantify changes in the shape of temperature distribution[51], as well as from distinct temporal- and spatial-scales being considered[52].

We also note that projections of compound hot extremes show increasingly large intermember/intermodel spread, which is markedly larger than that of daytime/nighttime event projections (Fig. 6). In light of our physical interpretations (Fig. 3) and other recent studies[53,54], this large spread may be linked to increasingly diverging projections of precipitation and resultant discrepancies in land–air interaction physics. So more trustworthy projections of compound hot extremes with reduced uncertainties, particularly at a regional scale, should be built on deeper mechanism understandings, including synoptic dynamics and local-to-regional surface energy balance as well as their responses to anthropogenic forcings[54]. At continental to global scales, both our statistical analysis (Fig. 2e, f) and some existing literature[16,31] strongly suggest that changes in synoptic dynamic–thermodynamic drivers are likely secondary to the direct radiative forcing of increasing GHGs in driving long-term changes in compound hot extremes.

## Methods

**Observations and simulations**. Gridded observations of near-surface Tmax and Tmin at a horizontal resolution of 3.75° longitude × 2.5° latitude are taken from the HadGHCND data set[22]. Considering the availability of observations for producing this data set, we focus our analysis on the Northern Hemisphere land areas. Only grid-boxes with no more than one missing value for Tmax/Tmin over 1960–2012 are used. The single missing value is infilled by the average of its neighboring 2 days' observations. To test the sensitiveness of trend estimates to the choice of data set, we also use daily Tmax and Tmin observations from the Berkeley Earth Surface Temperature data set[23], which are re-gridded onto 3.75° × 2.5° grids

following the HadGHCND's resolution and geography and then masked by the observation availability in the HadGHCND.

Historical simulations and projections of climate variables are taken from the Coupled Model Intercomparison Project Phase 5 (CMIP5)[55]. To improve the sampling of internal variability, each model used here is required to have at least three ensemble members with Tmax/Tmin outputs available at a daily scale in each forced experiment, as detailed in Supplementary Table 1. Note that the experiments including both anthropogenic and natural forcings (ALL) end in 2005, after when the RCP4.5 simulations are employed to extend historical ALL-forcing simulations till 2012. Following the observation's resolution and geography, we apply a bilinear interpolation algorithm to re-grid model outputs onto the same 3.75° × 2.5° grid and then mask the re-gridded data by the observations.

For projections of population, we use spatially explicit global population scenarios[43] which account for both changes in the size and spatial distribution of future population. These projections are provided at a spatial resolution of 1/8° × 1/8° and at a decadal interval over 2010–2100. To reconcile the spatial resolution and availability of grids in climate and population projections, we compute 3.75° × 2.5° population grids by tallying up the total number of persons in those 1/8° population grids[42] included in the domain of each climate grid, and then mask them by the observation grids.

**Summertime hot extremes, frequency, and intensity**. A hot day/night is considered when Tmax/Tmin is higher than its historical 90th percentile for the specific calendar day during summer (June–August)[56]. Such daily-based 90th percentiles are determined by ranking historical (1960–2012) 15-day samples surrounding this day (7 days before and after, i.e., total samples 15 × 53 = 795 days). These daily-based percentiles are, on one hand, stronger than the seasonal-fixed threshold during peak summer, thus acting to distinguish especially intense events from more typical cases; on the other hand, slightly lower than seasonal-fixed threshold during early/late summer, thereby permitting to identify hot extremes at different stages of summer[56]. Thus, these daily-based percentiles take into account intra-seasonally varying preparedness and acclimatization potential of human and ecosystems against excess heat[56,57]. The adoption of daily-based percentiles also avoids possible inhomogeneity in frequency and intensity series of temperature extremes[58].

On this basis, we define three types of summertime hot extremes: a compound hot extreme—sequential occurrence of a hot day and a hot night within 24 h; an independent hot day—a hot day without a following hot night; and an independent hot night—a hot night without a preceding hot day.

The frequency for each type is the number of days satisfying corresponding constraints. The intensity is measured by the temperature exceedance(s) above corresponding threshold(s), thus highlighting the detrimental effects of excess heat above high background temperatures. We calculate the hemispheric-scale frequency and intensity of summertime hot extremes by averaging area-weighted grid values. We compute observed trends for frequency and intensity of summertime hot extremes and other physical variables using the nonparametric Theil–Sen's method[59,60] and estimate their 90% confidence interval based on the method proposed in ref. [61]. We perform the nonparametric Mann-Kendall test of the null hypothesis of trend for each grid at the 0.05 significance level[62,63]. Absolute trends (days decade$^{-1}$ for frequency and °C decade$^{-1}$ for intensity) are also converted to relative changes (% decade$^{-1}$ for both) with respect to their climatological means over 1961–1990, to facilitate inter-type comparisons (Supplementary Fig. 3).

**Roles of general warming and changing variability**. We first estimate the general warming signals by fitting a second-order polynomial to summer-mean Tmax/Tmin during 1960–2012 for each grid box. Then, with these general warming signals removed from daily Tmax/Tmin, the frequency and intensity are re-computed based on Tmax/Tmin residuals. The trends for these re-computed frequency and intensity are assumed to be dictated by evolving variabilities of summertime Tmax/Tmin (including interannual variability, seasonal cycle, intraseasonal, and diurnal variability). Accordingly, the remaining proportion in trends for original series is believed to be ascribed to the general warming (i.e., mean-state shift). The 5–95% uncertainty range of observed relative contributions is estimated through randomly sampling valid grid-boxes 100,000 times.

**Formal detection and attribution**. We employ an optimal fingerprinting method for the detection and attribution of observed changes in summertime hot extremes[38]. Observed changes (**Y**) are represented as a sum of scaled fingerprints (**X**) of various external drivers, plus internal climate variability (**ε**)

$$\mathbf{Y} = \mathbf{X}\beta + \mathbf{\varepsilon}. \tag{1}$$

The MME mean of forced simulations are used to construct the fingerprints, and outputs from pre-industrial control runs are used to estimate internal climate variability. These fingerprints, in both frequency and intensity, are then pre-processed into nonoverlapping 3-year-mean time series consisting of 18 data samples over 1960–2012. The anthropogenically forced signal (ANT) is represented as the difference between MME mean responses to ALL and to NAT (natural) forcings. Furthermore, the signal forced by other anthropogenic drivers (OANT,

dominated by aerosols and large-scale land-use changes[6]) is extracted from ANT by excluding the GHG-forced signal. The regression coefficients (scaling factors) $\boldsymbol{\beta}$ scale the fingerprints to best fit observed changes. The regression is resolved following the scheme proposed in ref. [38]

$$\tilde{\beta} = \left(\mathbf{X}^T \mathbf{C}_N^{-1} \mathbf{X}\right)^{-1} \mathbf{X}^T \mathbf{C}_N^{-1} \mathbf{Y}. \qquad (2)$$

To fit and test the regression models, we need two independent estimates for inversed covariance structure of the internal climate variability $\left(\mathbf{C}_N^{-1}\right)$. Specifically, we divide these pre-industrial control simulations into 64 nonoverlapping chunks and then separate them into two sets, which are used for data pre-whitening and estimating the 5–95% uncertainty range of scaling factors $\tilde{\beta}$, respectively. We conduct a regularized estimate of the covariance matrix of internal climate variability[39], which yields a full rank covariance matrix and avoids the underestimation of the lowest eigenvalues occurring in the original covariance matrix.

If the scaling factor for specific external forcing excludes zero, the influence of this forcing is deemed detectable in observed changes. Furthermore, when the scaling factor contains the unity, we claim that the MME mean of forced responses is consistent with observations. If the scaling factor is smaller (larger) than one, the magnitude of responses to this forcing is overestimated (underestimated) in simulations compared with observations. To ensure the validity of detection and attribution analysis, a standard residual consistency test[38] is also implemented to evaluate models' performance in reproducing internal variability of the frequency and intensity of summertime hot extremes. All results shown pass this test at the 0.05 significance level. Based on a successful detection, attributable portion in observed trends for frequency and intensity are computed as the product of simulated linear trends for these indices and their respective scaling factors. The 5–95% uncertainty range for attributable changes is then obtained by multiplying the MME mean forced changes with corresponding scaling factors' uncertainty range.

**Observationally constrained projections**. The detection and attribution analysis provides an optimal estimate of the scaling to better match the simulated amplitude of forced changes to observed signals[40]. By exploiting this calibration effect on forced responses, we produce constrained projections of summertime hot extremes during 2013–2099 under RCP4.5 and RCP8.5. More specifically, we scale raw projections of frequency and intensity changes in response to various external forcings by multiplying corresponding scaling factors[40]. We note that such extension of simulations to future periods may introduce inhomogeneities in the frequency and intensity series (as revealed in ref. [58]). Such inhomogeneities, however, turn out to be negligibly small (Supplementary Fig. 12). For the historical period (1960–2012), we reconstruct simulated anomalies (relative to 1960–2012) of changes in hot extremes by summing optimally scaled MME mean responses to GHG, OANT, and NAT (via the three-signal detection). For the period after 2012, the MME mean responses under RCP4.5 and RCP8.5 are scaled by the scaling factor for ANT. Finally, we adjust the historical mean (1960–2012) of the reconstructed series to match the observed counterpart. Apparently, this observationally constrained projection method assumes the propagation of current biases of simulated forced changes into future, and does not account for errors exclusive to the future, such as a sudden shutdown in the thermohaline circulation[40].

**Specific levels of global warming**. Based on the re-gridded daily Tmax and Tmin outputs from CMIP5 models (Supplementary Table 1), we compute monthly anomalies (relative to 1861–1890) of daily mean surface air temperatures at each grid box for each simulation. Then, weighting the gridded values by the cosine of their latitudes, we calculate the ensemble mean annual global mean surface air temperature anomalies for individual models and average these ensemble means to obtain the MME mean global warming magnitudes. Similar to the methods of King et al.[64], we measure specific levels of global warming by decadal-average MME mean global warming magnitudes.

**Projection of population exposure to hot extremes**. Considering both population dynamics and hazard increases[42], our measure of population exposure refers to the number of person-days experiencing hot extremes, calculated as the summer number of events multiplied by the number of people exposed. The projected exposure, per decade, is computed from the spatial average of the product of decadal-average event frequency at each grid and the total population at that grid in that decade. Note that here we have to rely on raw projections of hot extremes instead of observationally constrained ones for hazard aspect in calculating exposure, since the latter projection scheme can not be performed on a grid-scale basis as methodologically required. Potential biases in estimating population exposures by using unconstrained projections of hazards are discussed in the main text.

Among various integrated scenarios constituted by RCPs and SSPs, we show a RCP4.5-SSP1 combination to frame a world evolving into a future with relatively low challenges to adaptation and mitigation, and a RCP8.5-SSP3 combination to characterize a world with rapid growth in emissions and populations, i.e., the most challenging scenario[65].

## Data availability

The observational data that support the findings are publicly available. The HadGHCND data are available at https://www.metoffice.gov.uk/hadobs/hadghcnd/. The Berkeley surface air temperature data are available at the Berkeley Earth website (http://berkeleyearth.org/). The CRU data could be accessed via http://www.cru.uea.ac.uk/data/. The NCEP-NCAR reanalysis could be gained through https://www.esrl.noaa.gov/psd/. The CMIP5 model outputs are accessible via the website (https://esgf-node.llnl.gov/projects/cmip5/). The spatially explicit global population projection data are publicly available at https://sedac.ciesin.columbia.edu/data/set/popdynamics-pop-projection-ssp-2010-2100/data-download.

## Code availability

The data in this study were analyzed with publicly available tool packages in MATLAB and the figures were produced with NCAR Command Language. All the scripts are available upon requests.

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

## Acknowledgements

We thank the Met Office Hadley Center, the Berkeley Earth project, the National Centers for Environmental Prediction, the National Center for Atmospheric Research, and the Climatic Research Unit for compiling the observational and reanalysis data sets and making them publicly available. We appreciate the Program for Climate Model Diagnosis and Intercomparison and the World Climate Research Programme's Working Group on Coupled Modeling for their contributions in producing the CMIP5 multi-model data. We also thank Dr. Bryan Jones and Dr. Brian C. O'Neill who developed and compiled the spatially explicit global population projections. J.W., Y.C., Z.Y. and P.Z. were jointly supported by the National Key Research and Development Program of China (Grant No. 2018YFC1507700) and the Strategic Priority Research Programme of the Chinese Academy of Sciences (Grant No. XDA20020201). J.F. acknowledges support from the National Key Research and Development Program of China (Grant No. 2016YFA0600403). S.F.B.T. was supported by the UK-China Research & Innovation Partnership Fund through the Met Office Climate Science for Service Partnership (CSSP) China as part of the Newton Fund.

## Author contributions

J.W., Y.C., and S.F.B.T. designed the research; J.W. carried out most calculations and result interpretations, created all figures and wrote the draft, with assistance from Y.C.; S.F.B.T. gave valuable comments on the analysis and helped with the writing and editing of the paper; Z.Y., P.Z., J.F., and J.X. took part in the discussion on the paper and contributed to the interpretation of the results.

## Competing interests

The authors declare no competing interests.
