## [Peer Review File · Nature Communications]

Reviewers' comments:

Reviewer #1 (Remarks to the Author):

Review of "Anthropogenically-driven increases in the risks of summertime compound hot extremes" by Wang et al.

This study seeks to quantify the change not only in extreme hot days or nights but "compound hot extremes" which are a combination of hot daytime and nighttime temperatures. A larger observed warming signal in compound hot extremes is found than for extreme hot nights and days separately. An optimal fingerprinting technique is applied to attribute the trend in compound hot extremes to the anthropogenic influence on the climate (through greenhouse gas emissions).

Overall the study is useful and the methodology is sound. The analysis is well presented in the Figures and the manuscript is generally well written. I think the authors have overstated the novelty of their analysis (as I will explain below) but aside from that I don't have any major issues with the study.

Main comment:

The background to the analysis correctly states that combined daytime and nighttime heat has pronounced impacts and cites relevant studies. However, the authors suggest that little previous work has been performed using day and night temperatures in combination. There have been several studies using indices like Excess Heat Factor (EHF; Nairn & Fawcett, 2014) which use mean temperatures, i.e. $(T_{\max} + T_{\min})/2$ to examine heatwave changes (e.g. Perkins et al., 2012; Perkins & Alexander, 2013). It is worth using the heatwave literature more to contextualise your work, which is still novel but not to the extent it is currently stated to be.

Minor comments:

L77: "of mid" to "of the mid"

L81-83: Sentence starting "Thus, selecting...." I'm not sure what you mean here. I think as long as a result based on a univariate index is clearly framed there isn't any issue.

L87-88: The negative trends in central-eastern China may be due to aerosol increases the effects of which may be under-represented in some models.

L92-104: There is literature around change in the shape of daily temperature distributions (e.g. Donat & Alexander, 2012), although there is some disagreement between studies. A brief discussion in the context of your analysis of temperature variability may be warranted.

L276-277: It may be worth noting that other studies also use decadal-mean global-mean temperatures from projections when defining specific global warming levels (e.g. King et al., 2017).

Figure 4b: I suggest using colours with greater contrast to make the Figures easier to read. Additional labels on the graph may also help.

References

Donat, M. G., & Alexander, L. V. (2012). The shifting probability distribution of global daytime and night-time temperatures. *Geophysical Research Letters*, 39(14), n/a-n/a. <https://doi.org/10.1029/2012GL052459>

King, A. D., Karoly, D. J., & Henley, B. J. (2017). Australian climate extremes at 1.5 °c and 2 °c of global warming. *Nature Climate Change*, 7(6). <https://doi.org/10.1038/nclimate3296>

Nairn, J., & Fawcett, R. (2014). The Excess Heat Factor: A Metric for Heatwave Intensity and Its

Use in Classifying Heatwave Severity. *International Journal of Environmental Research and Public Health*, 12(1), 227–253. <https://doi.org/10.3390/ijerph120100227>

Perkins, S. E., & Alexander, L. V. (2013). On the Measurement of Heat Waves. *Journal of Climate*, 26(13), 4500–4517. <https://doi.org/10.1175/JCLI-D-12-00383.1>

Perkins, S. E., Alexander, L. V., & Nairn, J. R. (2012). Increasing frequency, intensity and duration of observed global heatwaves and warm spells. *Doi.Org*, (20). <https://doi.org/10.1029/2012gl053361>

Reviewer #2 (Remarks to the Author):

The authors find that the frequency and intensity of compound hot extremes (day+night) are increasing faster than hot days or hot nights alone. This result is shown to be driven by summertime mean warming, and is attributed to human GHG emissions. While the result is interesting, I'm not able to recommend this paper for publication in its current form – if the authors were able to show the physical drivers behind the result, or more quantitatively demonstrate why we should care about compound hot extremes more than just hot days or nights, then this paper could be appropriate for *Nature Communications*.

1. Why is this NH only? HadGHCND provides some SH data, ie Australia, South Africa, parts of SE S. America. The authors should do this analysis globally.

2. I am not clear what we learn via the detection and attribution analysis. The authors show that most of the compound hot extremes change is due to summer mean warming, and there have been many studies attributing >100% of that summer mean warming to human GHGs. I don't see any reason we would expect a different result for compound hot extremes, and indeed the authors find >100% of the change due to anthropogenic forcing. The authors should either explain what is novel/interesting about this analysis or make it a less prominent part of the paper.

3. There is no discussion or analysis of why compound hot extremes increase more than hot days or nights. Is it just a statistical result of mean warming? Are there different dynamics / land-air interactions on compound hot days? For this paper to be considered in a journal like *Nature Communications*, I think a physical explanation of this result is necessary.

4. The authors state that compound hot days are more impactful than hot days or hot nights. Is this true outside of human health? The authors need to justify more clearly and (preferably quantitatively) why compound hot days/nights are bad, and how much worse they are than hot days and nights alone.

5. Why are only 4 CMIP5 models used? Are there really only 4 models that provide all the data the authors use? All available models should be used in this kind of analysis.

Fig 1, 2: Please use a diverging color map so near-zero values can be identified. As it is now zero is yellow and is very hard to distinguish from small positive values.

Fig 2e, f: If the horizontal lines are the MME values, they should only span the model results and not the OBS bars.

Fig 4a: The legend states that the red markers are "brown".

Fig 4b: Please explain this figure more in the caption – if one isn't familiar with detection and attribution, "scaling factor" is unclear and it's hard to tell from reading the main text and figure caption what this panel shows.

Fig 5: Add a legend showing which symbols correspond to which models.

Reviewer #3 (Remarks to the Author):

This paper takes a look at the changes in heat waves since 1960, and moving forward into the future. I like that the authors look at the 3 different types of heat waves, better. But I think the paper needs a lot of work, maybe too much work. The dataset they use for historical analysis is probably not suited to do so, and the CMIP5 models really don't do a very good job reproducing the past...so sometimes I have a hard time putting much stock in what they say about the future....especially at the surface level. These issues really need to be robustly addressed before this manuscript should be published.

Comments:

1. The HADGHCNd is not a homogenized dataset, unless I am mistaken. Which means it cannot really be used it to examine changes over time. The authors don't even discuss the dataset's limitations, origins, or the processing applied to it (underlying observations origin, QC, homogenization, infilling, etc.). This is a pretty major issue, can you please compare it to the Berkely Earth Surface Temperature dataset daily resolution version?
2. The authors calculate the linear trends via OLS regression. A lot of people use non-parametric methods to calculate linear trends, like Mann-Kendall's tau. I think the authors need to convince the readers that the sensitivity to linear trend methodology isn't overly large. The sensitivity isn't even recognized in the document as it is.
3. What about the sensitivity to start and end date? I have a fair amount of experience in this area and it's shown me that this sensitivity can be large. Have the authors looked at this sensitivity, and if so, can something be mentioned in the manuscript?
4. Lines 86-91. The authors failed to convince me that the multi-model ensemble means were able to reproduce the observed trends in any of the 6 metrics in Figure 1. The features that the author admits the models failed to reproduce were the only conspicuous features in the observations. Can the authors please quantify the similarity of the spatial patterns? Probably the same goes for figure S5....is there any way to quantifiably convince the reader that the simulation is able to reproduce the observed area averaged time series? Is this all based on Scaling factors? I'm feeling uneasy about the
5. Overall readability needs to be improved.
6. Was the calculation of percentiles done using empirically or with assumptions on the distribution (i.e. taking the 90th percentile as the 1.285std above the mean)?
7. I've never heard of observationally-constrained projections, and I really didn't understand it after reading your paper. "By exploiting the calibration effect of scaling factors on forced responses," is going to need to be expanded and some references provided so the reader is convinced that this is reasonable practice.
8. Lines 179-180. "the observational constraint amplifies the response of compound hot extremes to 180 anthropogenic forcing, compared to the raw simulations" – I'm feeling pretty uneasy about this, honestly. Does this mean the scaling factors are driving the conclusions? If that is the case I think the scaling factors aspect needs to be sold better.

REPLIES TO EDITOR'S AND REVIEWS' COMMENTS

We sincerely appreciate editor's and three referees' highly constructive suggestions and comments, upon which we have revised our manuscript. The revised version, we believe, now has become a better piece of work. Hope our revisions and responses are clear and satisfactory. If any further revision is required, we'd be happy to follow your suggestions again.

Point-by-point responses are presented as follows with referees' comments in black and our responses in blue. In the revised manuscript, all major revisions are highlighted in blue as well.

Referee #1

Introductory comment:

Review of "Anthropogenically-driven increases in the risks of summertime compound hot extremes" by Wang et al.

This study seeks to quantify the change not only in extreme hot days or nights but "compound hot extremes" which are a combination of hot daytime and nighttime temperatures. A larger observed warming signal in compound hot extremes is found than for extreme hot nights and days separately. An optimal fingerprinting technique is applied to attribute the trend in compound hot extremes to the anthropogenic influence on the climate (through greenhouse gas emissions).

Overall the study is useful and the methodology is sound. The analysis is well presented in the Figures and the manuscript is generally well written. I think the authors have overstated the novelty of their analysis (as I will explain below) but aside from that I don't have any major issues with the study.

Response to comment: Thank you for your positive and encouraging comments. We have revised our manuscript by fully accounting for your concerns, particularly that relevant to the statement about novelty.

(1) *The background to the analysis correctly states that combined daytime and nighttime heat has pronounced impacts and cites relevant studies. However, the authors suggest that little previous work has been performed using day and night temperatures in combination. There have been several studies using indices like Excess Heat Factor (EHF; Nairn & Fawcett, 2014) which use mean temperatures, i.e. $(T_{max} + T_{min})/2$ to examine heatwave changes (e.g. Perkins et al., 2012; Perkins & Alexander, 2013). It is worth using the heatwave literature more to contextualise your work, which is still novel but not to the extent it is currently stated to be.*

Response to comment: We agree with you that it is not reasonable to claim that few previous studies have been conducted using day and night temperatures in combination.

Changes made in response to comment: Following your suggestion, we now have better contextualized our work in existing literature as: "Some heatwave definitions accordingly consider both day and night temperatures, for instance using daily mean temperature as a measurement^{13,14}. However, compared to the widely-investigated univariate hot days and hot nights^{4,5,9,10}, current understandings about past and future changes in combined daytime-nighttime hot extremes as well as underlying drivers remain too limited to inform development of type-specific strategies for adaptation and mitigation."

Newly added references:

13. Perkins, S. E. & Alexander, L. V. On the measurement of heat waves. *J. Clim.* **26**, 4500–4517 (2013).

14. Nairn, J. R. & Fawcett, R. J. The excess heat factor: a metric for heatwave intensity and its use in classifying heatwave severity. *Int. J. Environ. Res. Public Health* **12**, 227–253 (2014).

(2) L77: "of mid" to "of the mid"

Response to comment: Your suggestion has been followed.

(3) L81-83: Sentence starting “Thus, selecting....” I’m not sure what you mean here. I think as long as a result based on a univariate index is clearly framed there isn’t any issue.

Response to comment: Yes, it is true that as long as a result based on a univariate index is clearly framed there is not any issue.

We intended to highlight that increases in compound hot extremes are of greater magnitude (in a percentage manner, % decade⁻¹) and higher significance than those in independent hot days (see **the following Figure R1**). In this context, the increases in intensity and frequency for traditional hot days or warm days (Tmax>=90th), which by our classification should consist of compound hot extremes (Tmax>=90th and Tmin>=90th) and independent hot days (Tmax>=90th and Tmin<90th), is lower than those for compound events but higher than those for independent hot days (see the figure below). Indeed, the number of grids seeing significant changes in compound hot extremes is markedly more than those for independent hot days. That’s to say, using trends for traditional hot days to represent changes in Tmax-related extremes may mask stronger and more significant signals related to compound hot extremes as a sub-category. That also underpins the rationale of anatomizing traditional hot days and nights into three sub-types.

Changes made in response to comment: To deliver this point more clearly, this sentence has changed into: “Thus, previous estimates of traditionally-defined warm days’ trends, which reflect mixed signals of changes in independent hot days and compound events, may under-represent faster and more significant increases in compound hot extremes as a sub-type of Tmax-related hot extremes (Fig. S3 a-d)”

Figure R1 | Linear trends for percentage changes (relative to 1961-1990) in frequency and intensity of summertime T_{max} -related hot extremes during 1960-2012. a, b, Compound hot extremes; c, d, Independent hot days; e, f, Traditionally-defined warm days ($T_{max} \geq 90^{\text{th}}$ percentile). Stippled regions indicate the significance (at the 95% confidence level) of long-term trend.

(4) L87-88: *The negative trends in central-eastern China may be due to aerosol increases the effects of which may be under-represented in some models.*

Response to comment: Yes, following your suggestion, the effects of increasing aerosol and under-representation in models have been mentioned in the revised main text.

(5) L92-104: *There is literature around change in the shape of daily temperature distributions (e.g. Donat & Alexander, 2012), although there is some disagreement between studies. A brief discussion in the context of your analysis of temperature variability may be warranted.*

Response to comment: We appreciate this kind suggestion. Indeed, a brief discussion about the changes in temperature variability is relevant to our analysis.

Changes made in response to comment: Following your suggestion, we have added some discussions about previous literature on changes in temperature variability in the Discussion Section.

(6) L276-277: *It may be worth noting that other studies also use decadal-mean global-mean temperatures from projections when defining specific global warming levels (e.g. King et al., 2017).*

Response to comment: We thank the referee for this information.

Changes made in response to comment: We have added this information in the revised manuscript.

(7) *Figure 4b: I suggest using colours with greater contrast to make the Figures easier to read. Additional labels on the graph may also help.*

Response to comment: Your suggestion has been followed. We have modified the color scheme and added labels in this Figure in the revised manuscript.

Figure 5 | Scaling factors and attributable changes. **a**, Best estimate (cross) and uncertainty range (bars) of scaling factors for ANT (orange) and NAT (blue). **b**, as a but for GHG (purple), OANT (green), and NAT (blue) in three-signal detection analysis. **c**, The best estimates (shaded bars) and uncertainty ranges (black bars) in observed changes (gray) of summertime compound hot extremes and those attributable only to GHG (purple), OANT (green) and NAT (blue) from three-signal analysis. The meaning of the scaling factors and the calculation of attributable changes see Methods-‘Detection and attribution’ section. All

the uncertainty ranges refer to the 5-95% interval.

References

Donat, M. G., & Alexander, L. V. (2012). The shifting probability distribution of global daytime and night-time temperatures. *Geophysical Research Letters*, 39(14), n/a-n/a. <https://doi.org/10.1029/2012GL052459>

King, A. D., Karoly, D. J., & Henley, B. J. (2017). Australian climate extremes at 1.5 °c and 2 °c of global warming. *Nature Climate Change*, 7(6). <https://doi.org/10.1038/nclimate3296>

Nairn, J., & Fawcett, R. (2014). The Excess Heat Factor: A Metric for Heatwave Intensity and Its Use in Classifying Heatwave Severity. *International Journal of Environmental Research and Public Health*, 12(1), 227–253. <https://doi.org/10.3390/ijerph120100227>

Perkins, S. E., & Alexander, L. V. (2013). On the Measurement of Heat Waves. *Journal of Climate*, 26(13), 4500–4517. <https://doi.org/10.1175/JCLI-D-12-00383.1>

Perkins, S. E., Alexander, L. V., & Nairn, J. R. (2012). Increasing frequency, intensity and duration of observed global heatwaves and warm spells. *Doi.Org*, (20). <https://doi.org/10.1029/2012gl053361>

Response to comment: Thank you for providing these relevant and useful references, which have been cited now.

Referee #2

Introductory comment:

The authors find that the frequency and intensity of compound hot extremes (day+night) are increasing faster than hot days or hot nights alone. This result is shown to be driven by summertime mean warming, and is attributed to human GHG emissions. While the result is interesting, I'm not able to recommend this paper for publication in its current form – if the authors were able to show the physical drivers behind the result, or more quantitatively demonstrate why we should care about compound hot extremes more than just hot days or nights, then this paper could be appropriate for Nature Communications.

Response to comment: Thank you very much for these positive comments and constructive suggestions, which help improving our manuscript a lot. We have exerted our best to address your concerns. In particular, we have added the analysis and discussion about physical processes determining the spatial pattern of compound event trends and their faster increases compared to other types, and also have presented more evidences to underpin the rationale for caring more about compound hot extremes. Detailed point-by-point responses are presented as follows.

(1) *Why is this NH only? HadGHCND provides some SH data, ie Australia, South Africa, parts of SE S. America.*

The authors should do this analysis globally.

Response to comment: We focus on the Northern Hemisphere because the data availability in nearly the entire Southern Hemisphere (except in some parts of Australia) fails to satisfy the criterion that only grid-boxes with no missing values for Tmax/Tmin over 1960-2012 are used for analyses. Hence, it is reluctant and improper to claim that we can do analysis quasi-globally by involving the sparse data in the Southern Hemisphere, though quite a few studies did so. As you can see, precise observational results are the premise for the follow-up detection and attribution analysis as well as for constraining future

projections. Such consideration motivates us to deliver observation-based information as exactly as we can to help readers understanding the following results.

So in light of temporal availability and spatial coverage of observational data, we prefer to focus on the Northern Hemisphere continents. As a matter of fact, our initial analysis did incorporate available data in Australia, finding that such an inclusion would not influence our conclusions in a significant manner.

Changes made in response to comment: A brief discussion about data gaps has been added in the Discussion Section.

(2) I am not clear what we learn via the detection and attribution analysis. The authors show that most of the compound hot extremes change is due to summer mean warming, and there have been many studies attributing >100% of that summer mean warming to human GHGs. I don't see any reason we would expect a different result for compound hot extremes, and indeed the authors find >100% of the change due to anthropogenic forcing. The authors should either explain what is novel/interesting about this analysis or make it a less prominent part of the paper.

Response to comment: Yes, it is not quite surprising to find that changes in compound hot extremes are largely attributable to anthropogenic forcing. The manuscript follows a basic storyline of observed changes, cause analysis, detection and attribution, observationally-constrained projections. The part of detection and attribution actually serves as an essential nexus between the observational part and the projection part, so it is very important for the structure and integrity of the analysis.

The rationale for conducting the detection and attribution analysis is documented as follows:

1-- There are some contrasting evidences indicating that the general warming versus changing variability is the main driver for changes in temperature extremes (Schär et al. 2004; Stott et al. 2004; Perkins-Kirkpatrick et al. 2017). But all existing analyses focused on univariate hot days or nights. Relative

contributions from these two changing properties of temperature distributions to observed changes in compound hot extremes remain unknown. That motivates us to answer this question in the first instance. Based on established relationship between the general warming and anthropogenic forcing, we indeed could speculate the dominant role of anthropogenic warming. But apparently, such speculation is qualitative. General readers and policy-makers would require exact quantification (i.e. the values) of respective contributions from various natural and anthropogenic forcings (e.g. GHG, anthropogenic and volcanic aerosols, solar activities) instead of GHG alone. This is particularly the case for hemispheric to global scale changes, which are largely driven by external forcings (Allen and Tett 1999; Allen et al. 2000). The quantification is therefore essential to inform people of the extent to which and how human activities have caused and will cause changes in hot extremes. For instance, both our results and other studies imply air cleaning actions in the future may result in extra increases in hot extremes (Wang et al. 2017; Xu et al. 2018). This also matters for future mitigation and adaptation against hot extremes. Without formal detection and attribution analysis, we can hardly provide relevant quantitative clues.

2-- Compared to observed changes, policy-makers may concern more about future changes in summertime hot extremes, as this information is crucial to design mitigation and adaptation strategies. However, due to substantial differences in models' climate sensitivities, the uncertainty range of projected changes of hot extremes remains too large to convince policy-makers to rely on relevant projections. As illustrated in this manuscript, results from detection and attribution analysis provide a useful constraint to calibrate modeled responses to external forcings, thus reducing the uncertainty sourced from climate sensitivity. This will lend more credibility to projected changes. So we would like to stress again that the detection and attribution is an indispensable link between observations and projections of high confidence.

Changes made in response to comment: We have clearly highlighted the rationale for conducting detection and attribution analysis in different sections of the revised manuscript.

Related References:

Schär, C., Vidale, P. L., Lüthi, D., Frei, C., Häberli, C., Liniger, M. A. & Appenzeller, C. The role of increasing temperature variability in European summer heatwaves. *Nature* **427**, 332–336 (2004).

Stott, P. A., Stone, D. A. & Allen, M. R. Human contribution to the European heatwave of 2003. *Nature* **432**, 610–614 (2004).

Perkins-Kirkpatrick, S. E., Fischer, E. M., Angélil, O. & Gibson, P. B. The influence of internal climate variability on heatwave frequency trends. *Environ. Res. Lett.* **12**, 044005 (2017).

Allen, M. R. & Tett, S. F. B. Checking for model consistency in optimal fingerprinting. *Clim. Dyn.* **15**, 419–434 (1999).

Allen, M. R., Stott, P. A., Mitchell, J. F. B., Schnur, R. & Delworth, T. L. Quantifying the uncertainty in forecasts of anthropogenic climate change. *Nature*. **407**, 617–620 (2000).

Wang, Z., Lin, L., Zhang, X., Zhang, H., Liu, L. & Xu, Y. Scenario dependence of future changes in climate extremes under 1.5 °C and 2 °C global warming. *Sci. Rep.* **7**, 46432 (2017).

Xu, Y., Lamarque, J. F. & Sanderson, B. M. The importance of aerosol scenarios in projections of future heat extremes. *Clim Change* **146**, 393–406 (2018).

(3) *There is no discussion or analysis of why compound hot extremes increase more than hot days or nights. Is it just a statistical result of mean warming? Are there different dynamics/land-air interactions on compound hot days? For this paper to be considered in a journal like Nature Communications, I think a physical explanation of this result is necessary.*

Response to comment: Thank you very much for this highly-constructive yet challenging comment, which substantially deepens our understanding about physical processes behind the attribution statement. There are several points that we would like to deliver firstly.

1. The statistical result of the dominant role of the summer-mean warming has clear physical implications that the continuously-accumulated heat in the atmosphere could largely determine frequency increase and intensification of compound hot extremes even with weaker and/or less frequent typical physical setups. This is supported by some quantitative analysis (Horton et al. 2015; Purich et al. 2014).

2. Synoptic dynamics and local land-air interactions are very important in determining the spatial patterns of trends for compound hot extremes and the difference in increasing rates amongst three types, as detailed below. But for long-term changes at a spatially-aggregated scale (i.e. hemispheric-scale in our case), external forcings are believed to be the main driver.

Enlightened by your comment, we now better frame the question as '*how dynamics (circulation) and thermodynamics (land-air interactions) explain the spatial heterogeneity of trends for compound hot extremes and differing increasing rates amongst three types*'. To this end, we divide the NH continents into 20 zones, and further examine the dependence of regional-scale changes in compound hot extremes on regional physical processes (Fig. 3)

In light of dynamics, anticyclonic conditions give rise to greater adiabatic heating, reductions in cloud and greater absorbed solar radiation. These physical processes, on one hand, directly lead to higher T_{max}; on the other hand, store more heat near the surface for subsequent release at night, thus partly offsetting the radiative cooling and elevating T_{min}. An increase in anti-cyclonic conditions should dynamically result in an increase in compound hot extremes. To prove this, we compute linear trends in summer-mean (i.e., June, July, and August) sea level pressures and 500-hPa geopotential heights to approximately represent anticyclonic activities, accounting for both influences of internal variability and anthropogenic warming on circulation changes (Horton et al. 2015; Swain et al. 2016). We find a significant relationship (Fig. 3b-c) that stronger increases in anticyclonic conditions correspond to larger increases in the frequency of compound hot extremes (compare Fig. S6 with Fig. 1). This is particularly pronounced in Europe, western Asia, southeastern Greenland and northeastern Asia, consistent with

previously-reported significant increases in synoptic-scale anticyclones there (Horton et al. 2015; Lee et al. 2017). This relationship is more significant using 500hPa height trends (Fig. 3b-c). Considering strong influences of the general warming on 500hPa height increases, the evidence that changes in compound hot extremes have been markedly influenced by changes in anticyclonic conditions seems not as strong as theoretically assumed.

From the perspective of land-air interaction, some studies have document influences of drying soil on spatial-temporal patterns of extreme Tmax via land-air feedback processes (e.g. Fischer et al. 2007; Seneviratne et al. 2010; Vogel et al. 2017; Donat et al. 2018). Other studies further point out that the drying soil and triggered land-air interactions are also crucial for the persistence of high temperature into night (e.g. Black et al. 2004; Miralles et al. 2014). Simply put, atop drying soil moisture, ground heat storage accumulates during daytime under clear sky conditions associated with anticyclonic cells, and strong ground heat fluxes released at night act to offset radiative cooling. This process is quite efficient in producing nighttime hot extremes, since the nocturnal boundary layer is normally shallow. That points to the potential of strong nocturnal land-air interaction in strengthening the coupling between daytime and nighttime heat (Cowan et al. 2017; Freychet et al. 2017; Russo et al. 2019). A large fraction of compound hot extremes indeed concur with drying soil and strong daytime&nighttime land-air interaction processes (Black et al. 2004; Fischer et al. 2007; Miralles et al. 2014; Nairn and Fawcett 2013).

In this regard, we use the correlation between detrended precipitation (generally representing soil moisture condition) and detrended temperatures (Tmax & Tmin) to measure the strength of land-air temperature coupling, as did in previous studies (Muller and Seneviratne 2012; Zscheischler and Seneviratne 2017). Fig. S6 c-d clearly indicates the global prevalence of anti-correlation between temperature and precipitation. Of note is that increases in compound hot extremes are found greater in the areas characteristic of stronger nocturnal land-air interactions (compare Fig. S6c with Fig. 1a). Fig. 3d confirms the statistical significance of such a physical linkage. By contrast, stronger daytime land-air

interaction is not necessarily translated into greater increases in compound hot extremes (Fig. 3e). This could be interpreted as that given strong daytime land-air interaction alone, resultant extreme Tmax has less chance to couple with an extreme Tmin, unfavorable to the occurrence of compound hot extremes (Fig. 3e).

Notably, stronger nocturnal land-air interactions overlap with greater increases in anticyclonic activities in some regions (Fig. 3 b-d, red and green symbols). Such a coincidence, along with above physical understandings, motivates us to infer that the combination of these two physical processes tends to strengthen the coupling between daytime and nighttime hot extremes (Fig. S7), leading to greater increases in compound events than decoupled hot days/nights.

We also realize that these physical drivers may also show responses to radiative forcings from anthropogenic emissions of greenhouse gases, i.e. the general warming. As mentioned above, Horton et al. (2015) and Lee et al. (2017) report that as the globe warms, synoptic-scale anticyclonic conditions have exhibited strong and significant increasing trends in Europe, western Asia and northeastern Asia (we didn't repeat their analysis). Regions typical of wet-dry transitional climates have expanded along with enhanced land-air interaction in response to the past global warming (Huang et al. 2016). These responses in physical drivers may enhance the coupling between coupled daytime and nighttime hot extremes, further elevating the probability of compound hot extremes. For future periods, there are strong evidences indicating that current land-air interaction hotspots may see even stronger soil moisture-air temperature interactions (Vogel et al. 2017; Donat et al. 2018; King 2019). Also, the ongoing dryland expansion may facilitate more land areas in the Northern Hemisphere to evolve into dry-wet transitional zones (Vogel et al. 2017; Huang et al. 2017). That may partly contribute to faster increases in frequency and intensity of compound hot extremes in the future (Fig. 6).

Changes made in response to comment:

The major points (1&2) have been clearly stated in the revised manuscript, along with above analysis

summarized and relevant figures added. Additionally, the limitation of the diagnostic method used here and the avenue for future studies in these regards are discussed in the Discussion Section.

Figure 3 | Scatter plots of the statistical relationship between changes in frequency of compound hot extreme and dynamic-thermodynamic physical drivers. a Classified climate zones and their acronyms. **b, c** Scatter plot between trends for circulation changes with (b) for sea level pressure and (c)

for 500-hPa geopotential heights and frequency trends for compound hot extreme averaged in twenty classified zones during 1960-2012. **d**, **e**, as in **b**, but for the plot between temperature-precipitation correlation with daily minimum (**d**) & maximum (**e**) and trends for frequency of compound hot extremes during 1960-2012. Before calculating correlation coefficients, both temperature and precipitation series are linearly detrended in (**d**) and (**e**). Each symbol represents one classified zone. Long and short dashed lines show the 95% confidence and prediction intervals of the regression equation, respectively. The linear regression equation, the proportion of the variance of Y explained by X (R^2), the Pearson correlation coefficient (*corr*), and its *p*-value (*P*) are shown in on each panel. For the calculation details for (b) and (c) see Text-S2.

Figure S6 | Linear trends for summer-mean sea level pressure (a) and 500hPa geopotential heights (b) based on the NCEP-NCAR R1 reanalysis, and correlation between CRU summer-mean daily maximum (c) /minimum temperature (d) and precipitation during 1960-2012. For the method for trend estimate in (a) and (b) see Text-S2. Before calculating correlation coefficients in (c) and (d), both maximum/minimum temperature and precipitation at each grid are linearly detrended.

Figure S7 | Distribution of observed frequency trends for summertime compound hot extremes in a dynamic-thermodynamic space, with observed trends for sea level pressures (a; SLP, x-axis), 500-hPa geopotential heights (b; HGT, x-axis) and correlations between daily minimum temperature and precipitation (CORR, y-axis) binned into intervals of 0.3 mb decade⁻¹(SLP), 2.5gpm decade⁻¹ (HGT) and 0.1 (CORR) respectively.

The color of each square here indicates the mean frequency trend averaged amongst those grids with their SLPs/HGTs and CORRs falling into corresponding bins. This is a synthesis plot for those one-dimensional plots as presented in Fig. 3 in the main text.

Related References:

Purich, A. *et al.* Atmospheric and oceanic conditions associated with southern Australian heat waves: a CMIP5 analysis. *J. Clim.* **27**, 7807-7829 (2014).

Cowan, T., Hegerl, G. C., Colfescu, I., Ballasina, M., Purich, A. & Boschat, G. Factors contributing to record-breaking heat waves over the Great Plains during the 1930s Dust Bowl. *J. Clim.* **30**, 2437–2461 (2017).

Freychet, N., Tett, S., Wang, J. & Hegerl, G. Summer heat waves over Eastern China: dynamical processes

and trend attribution. *Environ. Res. Lett.* **12**, 024015 (2017).

Swain, D. L., Horton, D. E., Singh, D., *et al.* Trends in atmospheric patterns conducive to seasonal precipitation and temperature extremes in California. *Sci. Adv.* **2**, e1501344 (2016).

Horton, D. E., Johnson, N. C., Singh, D., *et al.* Contribution of changes in atmospheric circulation patterns to extreme temperature trends. *Nature* **522**, 465–469 (2015).

Lee, M. H., Lee, S., Song, H. J. & Ho, C. H. The recent increase in the occurrence of a boreal summer teleconnection and its relationship with temperature extremes. *J. Clim.* **30**, 7493–7504 (2017).

Fischer, E. M., Seneviratne, S. I., Lüthi, D. & Schär, C. Contribution of land-atmosphere coupling to recent European summer heat waves. *Geophys. Res. Lett.* **34**, L06707 (2007).

Seneviratne, S. I., Corti, T., Davin, E. L., Hirschi, M., Jaeger, E. B., Lehner, I., Orlowsky, B. & Teuling, A. J. Investigating soil moisture–climate interactions in a changing climate: A review. *Earth-Science Rev.* **99**, 125–161 (2010).

Vogel, M. M., Orth, R., Cheruy, F., Hagemann, S., Lorenz, R., van den Hurk, B. J. J. M. & Seneviratne, S. I. Regional amplification of projected changes in extreme temperatures strongly controlled by soil moisture-temperature feedbacks. *Geophys. Res. Lett.* **44**, 1511–1519 (2017).

Donat, M. G., Pitman, A. J. & Angéllil, O. Understanding and reducing future uncertainty in mid-latitude daily heat extremes via land surface feedback constraints. *Geophys. Res. Lett.* **45**, 10,627–10,636 (2018).

Seneviratne, S. I., Lüthi, D., Litschi, M. & Schär, C. Land-atmosphere coupling and climate change in Europe. *Nature* **443**, 205–209 (2006).

Black, E., Blackburn, M., Harrison, G., Hoskins, B. & Methven, J. Factors contributing to the summer 2003 European heatwave. *Weather* **59**, 217–223 (2004).

Miralles, D. G., Teuling, A. J., van Heerwaarden, C. C. & de Arellano, J. V. Mega-heatwave temperatures due to combined soil desiccation and atmospheric heat accumulation. *Nature Geosci.* **7**, 345–349 (2014).

Russo, S., Sillmann, J., Sippel, S., Barcikowska, M. J., Ghisetti, C., Smid, M. & O'Neill, B. Half a degree and

rapid socioeconomic development matter for heatwave risk. *Nature Comm.* **10**, 136 (2019).

Nairn, J. R. & Fawcett, R. G. Defining heatwaves: heatwave defined as a heat-impact event servicing all community and business sectors in Australia. The Centre for Australian Weather and Climate Research (2013).

Mueller, B. & Seneviratne, S. I. Hot days induced by precipitation deficits at the global scale. *PNAS* **109**, 12398–12403 (2012).

Zscheischler, J. & Seneviratne, S. I. Dependence of drivers affects risks associated with compound events. *Sci. Adv.* **3**, e1700263 (2017).

Huang, J., Yu, H., Guan, X., Wang, G. & Guo, R. Accelerated dryland expansion under climate change. *Nature Clim. Change* **6**, 166–171 (2016).

King, A. D. The drivers of nonlinear local temperature change under global warming. *Environ. Res. Lett.* **14**, 064005 (2019).

Huang, J., Yu, H., Dai, A., Wei, Y. & Kang, L. Drylands face potential threat under 2°C global warming target. *Nature Clim. Change* **7**, 417–422 (2017).

(4) *The authors state that compound hot days are more impactful than hot days or hot nights. Is this true outside of human health? The authors need to justify more clearly and (preferably quantitatively) why compound hot days/nights are bad, and how much worse they are than hot days and nights alone.*

Response to comment: Thank you for this constructive comment. Based on case studies, a large body of existing literature reported compound hot extremes are more damaging to human health than individual daytime or nighttime hot weather, such as some deadly cases in Chicago heat wave in 1995 (Karl and Knight 1997), European heat wave in 2003 (Trigo et al. 2005; Grize et al. 2005; Charpentier 2011) and Russia heat wave in 2010 (Grumm 2011; Miralles et al. 2014). This is because (i) nighttime heat following a hot day accumulates body thermal load lethal to vulnerable people; (ii) people tend to keep a

high alert for daytime hot with proper precautions and adaptations adopted, but vigilance against continuing heat at night becomes loose because of a reasonable expectation of diurnal cycle of temperature based on their life experience, i.e. hot day and cool night (Nicholls et al. 2008; D'Ippoliti et al. 2010; Nairn and Fawcett 2013; Davis et al. 2016; Vaidyanathan et al. 2016).

We also agree with you that quantitatively distinguishing impacts from three types of hot extremes, regardless of on human health or other sectors, will add great value to this study. Some studies have documented influences of definitions of hot extremes on their relationship with morbidity and mortality. But, to our knowledge, there is still no study to date quantifying morbidity and mortality associated with three types of hot extremes as defined by us. Such quantification requires multi-decade daily mortality and morbidity data covering large parts of the Northern Hemisphere. We found a candidate data set of daily mortality on 500 cities/communities from 29 countries (<http://mccstudy.lshtm.ac.uk/>) fitting well for our purpose. However, the data producer replies to me in email that “Unfortunately, the data are only shared internally to the network following a specific protocol, and I am not at liberty to share it externally”. Even for regional-scale data like in China, our Chinese colleagues working on human health also replied to us that these data are not publicly available, i.e. confidential to us even for research purpose. It seems that a deeper collaboration with epidemiological community is badly needed to fill the gap! So we have to say we can't quantify and compare impacts of these three types of hot extremes on human health in the current study. But still, your valuable suggestion enlightens us a lot. When we can get access to these human health data, follow-up studies will be undertaken.

Similar limitation/restriction in data availability in other sectors (e.g. agriculture and electric power industry) hinders our quantitative analysis of impacts related to these summertime hot extremes outside the field of human health. To our knowledge, there are no studies to date investigating influences of these three types of hot extremes on ecology and agriculture. But, added impacts of nighttime hot extremes sequential to a daytime one on agriculture yields and ecosystem production could be inferred

from relevant studies (Peng et al. 2004; Lobell and Field 2007; Lobell and Gourdji 2012; Bhatt et al. 2014; Potopová et al. 2017). These added impacts are ascribed to nocturnal hot-triggered increases in the respiration cost without a potential benefit for photosynthesis, particularly during summer (Tan et al. 2015).

Changes made in response to comment: Emphasizing your suggestion, we have presented more evidences from existing case studies, still in a qualitative way, to underpin that compound hot extremes are more impactful than independent hot days/nights.

In the Discussion Section, we also clearly state the limitation of the current studies in quantitatively illustrating and distinguishing impacts from three types of hot extremes. Gaps, future avenues and emerging topics building on this study are also alluded to. These information may shed some light on future studies in filling above gaps.

Related References:

Karl, T. R. & Knight, R. W. The 1995 Chicago heat wave: How likely is a recurrence? *Bull. Amer. Meteorol. Soc.* **78**, 1107–1120 (1997).

Trigo, R. M., García-Herrera, R., Díaz, J., Trigo, I. F. & Valente, M. A. How exceptional was the early August 2003 heatwave in France? *Geophys. Res. Lett.* **32**, L10701 (2005).

Grize, L., Huss, A., Thommen, O., Schindler, C. & Braun-Fahrländer, C. Heat wave 2003 and mortality in Switzerland. *Swiss Med. Wkly.* **135**, 200–205 (2005).

Charpentier, A. On the return period of the 2003 heat wave. *Clim. Change* **109**, 245–260 (2011).

Grumm, R. H. The central European and Russian heat event of July–August 2010. *Bull. Amer. Meteorol. Soc.* **92**, 1285–1296 (2011).

Miralles, D. G., Teuling, A. J., van Heerwaarden, C. C. & de Arellano, J. V. Mega-heatwave temperatures due to combined soil desiccation and atmospheric heat accumulation. *Nature Geosci.* **7**, 345–349 (2014).

Nicholls, N., Skinner, C., Loughnan, M. & Tapper, N. A simple heat alert system for Melbourne, Australia. *Int. J. Biometeorol.* **52**, 375–384 (2008).

D'Ippoliti, D. *et al.* The impact of heat waves on mortality in 9 European cities: results from the EuroHEAT project. *Environ. Health* **9**, 37 (2010).

Nairn, J. R. & Fawcett, R. G. Defining heatwaves: heatwave defined as a heat-impact event servicing all community and business sectors in Australia. *The Centre for Australian Weather and Climate Research* (2013).

Davis, R. E., Hondula, D. M. & Patel, A. P. Temperature observation time and type influence estimates of heat-related mortality in seven US cities. *Environ. Health Perspect.* **124**, 795–804 (2016).

Vaidyanathan, A., Kegler, S. R., Saha, S. S. & Mulholland, J. A. A statistical framework to evaluate extreme weather definitions from a health perspective: a demonstration based on extreme heat events. *Bull. Amer. Meteorol. Soc.* **97**, 1817–1830 (2016).

Peng, S. *et al.* Rice yields decline with higher night temperature from global warming. *PNAS* **101**, 9971–9975 (2004).

Lobell, D. B. & Field, C. B. Global scale climate–crop yield relationships and the impacts of recent warming. *Environ. Res. Lett.* **2**, 014002 (2007).

Lobell, D. B. & Gourdji, S. M. The influence of climate change on global crop productivity. *Plant Phys.* **160**, 1686–1697 (2012).

Bhatt, D., Maskey, S., Babel, M. S. *et al.* Climate trends and impacts on crop production in the Koshi River basin of Nepal. *Reg. Environ. Change* **14**, 1291–1301 (2014).

Potopová, V., Zahradníček, P., Štěpánek, P. *et al.* The impacts of key adverse weather events on the field - grown vegetable yield variability in the Czech Republic from 1961 to 2014. *Int. J. Climatol.* **37**, 1648–1664 (2017).

Tan, J., Piao, S., Chen, A. *et al.* Seasonally different response of photosynthetic activity to daytime and

night-time warming in the Northern Hemisphere. *Global Change Biology*, **21**, 377–387 (2015).

(5) Why are only 4 CMIP5 models used? Are there really only 4 models that provide all the data the authors use?

All available models should be used in this kind of analysis.

Response to comment: Actually, we used five CMIP5 models in this study (Table S1). To reduce the uncertainties associated with model initial condition (i.e. better sample internal variability), it is required that each model has at least three ensemble members with daily Tmax/Tmin outputs available for each forced experiment (e.g., ALL, NAT, and GHG), as well as a pre-industrial control simulation spanning at least 500 years. Those criteria sift out 5 models as employed (i.e. CanESM2, CNRM-CM5, CSIRO-Mk3-6-0, HadGEM2-ES, and IPSL-CM5A-LR).

(6) Fig 1, 2: Please use a diverging color map so near-zero values can be identified. As it is now zero is yellow and is very hard to distinguish from small positive values.

Response to comment: Your suggestion has been followed. It now looks better.

Figure 1 | Linear trends for frequency and intensity of summertime hot extremes during

1960-2012 based on the HadGHCND observations. a, b, Compound hot extremes; c, d, Independent hot days; e, f, Independent hot nights. Stippled regions indicate the significance (at the 95% confidence level) of long-term trends. Event classifications and indices constructions are detailed in Methods.

Figure 2 | Contributions from changing summer mean temperature and temperature variability to changes in compound hot extremes. a, b, Observed changes in the frequency and intensity of compound hot extremes due only to changes in summer mean temperature. c, d, Observed changes in the frequency and intensity of compound hot extremes due merely to changes in temperature variability. e, f, Observed and modeled ensemble median contributions from changes in summer mean temperature (orange bars) and temperature variability (blue bars) to the area-weighted mean frequency (e) and intensity (f) changes. The vertical black bars show the 5-95% uncertainty range of observed relative contributions. Gray diamonds and circles indicate contributions from individual simulations of each model, with their MME median values represented by orange and blue lines.

(7) Fig 2e, f: If the horizontal lines are the MME values, they should only span the model results and not the OBS bars.

Response to comment: Thank you for your kind reminding. This issue has been corrected in the revised manuscript.

(8) Fig 4a: The legend states that the red markers are “brown”.

Response to comment: “brown” has been changed to “orange” in the revised manuscript.

Figure 5 | Scaling factors and attributable changes. **a**, Best estimate (cross) and uncertainty range (bars) of scaling factors for ANT (orange) and NAT (blue). **b**, as a but for GHG (purple), OANT (green), and NAT (blue) in three-signal detection analysis. **c**, The best estimates (shaded bars) and uncertainty ranges (black bars) in observed changes (gray) of summertime compound hot extremes and those attributable only to GHG (purple), OANT (green) and NAT (blue) from three-signal analysis. The meaning of the scaling factors and the calculation of attributable changes see Methods-‘Detection and attribution’ section. All the uncertainty ranges refer to the 5-95% interval.

(9) Fig 4b: Please explain this figure more in the caption – if one isn’t familiar with detection and attribution, “scaling factor” is unclear and it’s hard to tell from reading the main text and figure caption what this panel shows.

Response to comment: Yes, we have expanded the legend by adding ‘The meaning of the scaling

factors and the calculation of attributable changes see Methods-‘Detection and attribution’ section’. Also, we explain the meaning of scaling factor in more detail in the Methods. Otherwise, putting these explanations in the figure legend makes it quite long.

(10) Fig 5: Add a legend showing which symbols correspond to which models.

Response to comment: Your suggestion has been followed. The legend specifying the models’ name has been added in the revised figure, also shown as follows for your convenience.

Figure 6 | Observationally-constrained projections of area-weighted mean frequency and intensity of summertime hot extremes across the Northern Hemisphere land areas. Time series of historical and MME mean projected frequency (a) and intensity (b) of summertime compound hot extremes (purple lines), independent hot days (blue lines), and independent hot nights (green lines) over 2013–2099 under the RCP4.5 scenario. c, d, Same as a, b, but under the RCP8.5 scenario. Shadings in corresponding colors enclose the 5-95% range of individual simulations for each type. Black symbols represent simulated decadal average global mean surface air temperature (GMST) anomalies (relative to

1861-1890, refer to right y-axis) from 5 used models, with their names specified by the legend in **b**. Red circles enclose MME simulations of decadal-average GMST anomalies, the average among which reaches specific global warming levels of 1.5°C, 2°C and 4°C. Two vertical dashed lines locate the year of 1990 and 2020, when transitions of the dominant type of hot extremes occur.

Referee #3

Introductory comment:

This paper takes a look at the changes in heat waves since 1960, and moving forward into the future. I like that the authors look at the 3 different types of heat waves, better. But I think the paper needs a lot of work, maybe too much work. The dataset they use for historical analysis is probably not suited to do so, and the CMIP5 models really don't do a very good job reproducing the past...so sometimes I have a hard time putting much stock in what they say about the future....especially at the surface level. These issues really need to be robustly addressed before this manuscript should be published.

Response to comment: We appreciate the referee's encouraging comments. We have revised our manuscript by fully accounting for your concerns. In particular, we have included a series of robustness tests to the data and method used in our study. Also, models' ability in reproducing past temporal-spatial patterns of changes in three types of hot extremes are evaluated, with potential sources of inconsistency between observations and simulations discussed as well.

(1) *The HadGHCND is not a homogenized dataset, unless I am mistaken. Which means it cannot really be used it to examine changes over time. The authors don't even discuss the dataset's limitations, origins, or the processing applied to it (underlying observations origin, QC, homogenization, infilling, etc.). This is a pretty major issue, can you please compare it to the Berkeley Earth Surface Temperature dataset daily resolution version?*

Response to comment: Yes, your suggestion has been completely followed. You are definitely right about that near-surface daily maximum and minimum temperature time series from HadGHCND are not formally homogenized. We also realize that the confidence for conclusions of climate change studies, on extremes in particular, relies heavily on high-quality observations. To evaluate potential influences of

data inhomogeneity in estimating trends of summertime compound hot extremes, we compare HadGHCND results with those based on the Berkeley Earth Surface Temperature dataset in the revised manuscript, following your suggestion.

Firstly, we apply a bilinear interpolation algorithm to re-grid the $1.0^\circ \times 1.0^\circ$ daily maximum and minimum temperatures from the Berkeley Earth Surface Temperature dataset onto the $3.75^\circ \times 2.5^\circ$ grids following the HadGHCND's resolution and geography, and then mask the re-gridded data by available grids of HadGHCND. Then, we use the Theil–Sen method to re-estimate the linear trends for frequency and intensity of summertime compound hot extremes during 1960-2012 based on the re-gridded Berkeley Earth Surface Temperature dataset. We compare the results from these two datasets and find minor differences.

As shown in Fig. S1, the spatial patterns for frequency and intensity changes in summertime compound hot extremes are basically consistent using these two datasets. The differences in estimated trends for frequency are very small over much of the Northern Hemisphere. In terms of the intensity change, these two datasets do show some differences in Alaska, Northern Canada, Europe and Western Russia, which are likely to be associated with different origins and availability of observing stations in these regions as well as with differing interpolation scheme^{22,23}. As most of these regions are situated at higher latitudes, the weights assigned to these grids are smaller than those to low-to-middle latitude regions when creating domain-averaged series. Therefore, as shown in Fig. S1, the time series of the area-weighted frequency and intensity of summertime compound hot extremes averaged over the Northern Hemisphere land areas, based on these two datasets, are nearly identical. Since the following detection and attribution as well as projection are all conducted based on domain-averaged series, using the Berkeley series instead would not induce measurable differences in our conclusion and quantification.

Changes made in response to comment:

1-- We have inserted a figure (Fig. S1) to show the consistency in estimated trends for summertime compound hot extremes by using the HadGHCND and Berkeley Earth Surface Temperature datasets.

2-- We have added a brief discussion about sensitivity of data choice in the revised manuscript.

3-- The observation origin, quality control, generation processes, and limitations of HadGHCND dataset are further discussed in the Discussion section.

Figure S1 | Linear trends for frequency and intensity of summertime compound hot extremes during 1960-2012 based on the HadGHCND observations (a, b), the Berkeley Earth Surface Temperature data set (c, d), and their differences (e, f, Berkeley minus HadGHCND). g,h show area-weighted mean (g) frequency and (h) intensity of summertime compound hot extremes over the Northern Hemisphere lands during 1960-2012 based on these two datasets.

(2) The authors calculate the linear trends via OLS regression. A lot of people use non-parametric methods to calculate linear trends, like Mann-Kendall's tau. I think the authors need to convince the readers that the sensitivity to linear trend methodology isn't overly large. The sensitivity isn't even recognized in the document as it is.

Response to comment: Thanks for the suggestion. To reduce the sensitivity of trend estimator to potential outliers in time series of extremes, we now have turned to apply the non-parametric Theil–Sen approach to estimate linear trends for various indices during 1960-2012 and use the Mann-Kendall trend test for the significance of changes in the revised manuscript. Relevant conclusions on detection and attribution still hold by employing this alternative trend estimation method, though some values for the estimated trend do differ.

(3) What about the sensitivity to start and end date? I have a fair amount of experience in this area and it's shown me that this sensitivity can be large. Have the authors looked at this sensitivity, and if so, can something be mentioned in the manuscript?

Response to comment: We agree with the referee that the estimated trends may be sensitive to the starting and ending dates. We choose the analysis period of 1960-2012 since the data coverage is reasonably good over most of the North Hemisphere land areas after 1960 and the simulations of CMIP5 models forced by various external forcings end in 2012. To address your concern, we also calculate trends for frequency and intensity of summertime compound hot extremes during two other periods over 1955-2012 and 1965-2012, respectively. As Fig. S2 shows, the high similarity in terms of magnitude, significance and location of hotspots indicate the sufficient insensitiveness of our trend estimation to the choice of periods.

The result about sensitive tests to the analysis period has been presented in the revised text.

Figure S2 | Linear trends for frequency and intensity of summertime compound hot extremes during the periods of 1955-2012 (a, b), 1960-2012 (c, d), and 1965-2012 (e, f) based on the HadGHCND observations.

(4) Lines 86-91. The authors failed to convince me that the multi-model ensemble means were able to reproduce the observed trends in any of the 6 metrics in Figure 1. The features that the author admits the models failed to reproduce were the only conspicuous features in the observations. Can the authors please quantify the similarity of the spatial patterns? Probably the same goes for figure S5....is there any way to quantifiably convince the reader that the simulation is able to reproduce the observed area averaged time series? Is this all based on Scaling factors? I'm feeling uneasy about this.

Response to comment: We thank the referee for this comment. Basically, the multi-model mean captures the general feature of past changes in three types of hot extremes, as well as temporal evolution of domain-averaged series (Fig. S4 and Fig. S12). We quantify the spatial correlation between observed and simulated changes in the 6 matrix presented in Fig. 1 as you requested (Fig. S4). The significance of spatial correlation (at the 0.01 significance level at least) illustrates the general consistency between them, although the correlation coefficient seems not that large due mainly to the large sample

size for correlation calculation, i.e. thousands of grids. Such general consistency could serve our purpose of attribution and projection, both of which focus on large-scale (i.e. hemispheric-scale) changes. At an individual grid level, we find that the differences between the observed and simulated trends reduce markedly after using the non-parametric trend estimator as you recommended. But we still can't count on the multi-model ensemble mean to exactly reproduce observed changes in properties of summertime hot extremes in every part of the Northern Hemisphere, for the following reasons:

1-- The multi-model ensemble mean tends to smooth most of the internal climate variability, which also explains reasonable proportions of observed changes in extremes (particularly for the interannual to multi-decadal variability). The suppression or even absence of this part of contribution in multi-model mean partly accounts for the discrepancy between simulations and observations.

2-- Climate models participating CMIP5 still have limited ability to capture some complex local-regional processes, such as aerosol-cloud-radiation interactions and some key local-scale land-air interactions, which affect surface air temperatures. Hence, the discrepancy between the observations and simulations in regions with heavy air pollution and substantial land-use changes seems inevitable in the current stage.

3-- Last but not the least, varying degrees of models' climate sensitivities also hinder model's reliability in reproducing observed trends. A multi-model ensemble mean can reduce this part of uncertainties to some extent, but cannot average out them thoroughly. The scaling factor derived via detection and attribution analysis could be used to calibrate models' biased responses to external forcings, thereby further reducing this part of discrepancy. Such calibration effect works well for large-scale changes as measured by hemisphere-averaged series, but helps little at local-regional scales.

In Fig. 4 and Fig. S9, we just show original series of observed changes and un-calibrated simulations in

response to different forcings. Their differences reduce markedly after the simulated series are subject to calibration by the scaling factor (Fig. S12). More specifically, the observational series largely fall within the 5%~95% range of calibrated simulations (compare Fig. 4 & Fig. S9 with Fig. S12), though simulated inter-annual variability is still much weaker than the observation counterpart.

Changes made in response to comment: We have clearly state the models' ability and deficiency in capturing some local to regional features, with potential reasons listed in the revised manuscript.

Figure S4 | Same as Figure 1 but based on the multi-model ensemble means (MME) from five CMIP5 models forced by ALL forcings. The spatial correlation coefficients between observed and MME-simulated trends along with corresponding p -values are marked at the lower-left corner.

(5) Overall readability needs to be improved.

Response to comment: The revised manuscript has been thoroughly polished again by our native-speaker co-author, Prof. Simon F. B. Tett.

(6) Was the calculation of percentiles done using empirically or with assumptions on the distribution (i.e. taking the 90th percentile as the 1.285std above the mean)?

Response to comment: The calculation of the 90th percentiles is done empirically, rather than using the assumptions on the temperature distribution. It is consistent with the conventional practice recommended by Expert Team on Climate Change Detection and Indices (ETCCDI).

Changes made in response to comment: We cite a relevant reference (i.e. Della-Marta et al. 2007) to inform readers of the methods for calculation the 90th percentile.

Newly Added Reference:

55. Della-Marta, P. M, Haylock, M. R., Luterbacher, J. & Wanner, H. Doubled length of western European summer heat waves since 1880. *J. Geophys. Res. Atmos.* **112**, E15103 (2007).

(7) I've never heard of observationally-constrained projections, and I really didn't understand it after reading your paper. "By exploiting the calibration effect of scaling factors on forced responses," is going to need to be expanded and some references provided so the reader is convinced that this is reasonable practice.

Response to comment: Yes, your suggestion has been followed. Surely, this method is not created by us. It was introduced and employed in some pioneer works by Allen et al (2000) and Stott and Kettleborough (2002).

The anterior fingerprint detection-attribution identifies under-responsive and over-responsive models to certain external forcing. On this basis, application of this scaling improves inter-model consistency with projections from under-responsive models being scaled up and predictions from over-responsive models scaled down to best match observed changes. Since the major intent of this scaling technique is to match forced-simulations' amplitude with the observed signal, we prefer to name it as 'observationally-constrained projection'.

Following your suggestion, we have expanded the method description and cited relevant references for the convenience of readers. Also, the limitation of this method is briefly mentioned.

Newly Added Reference:

40. Allen, M. R., Stott, P. A., Mitchell, J. F. B., Schnur, R. & Delworth, T. L. Quantifying the uncertainty in forecasts of anthropogenic climate change. *Nature* **407**, 617–620 (2000).

Stott, P. A. & Kettleborough, J. A. Origins and estimates of uncertainty in predictions of twenty-first century temperature rise. *Nature* 416(6882), 723-726 (2002).

(8) Lines 179-180. *“the observational constraint amplifies the response of compound hot extremes to anthropogenic forcing, compared to the raw simulations” – I’m feeling pretty uneasy about this, honestly. Does this mean the scaling factors are driving the conclusions? If that is the case I think the scaling factors aspect needs to be sold better.*

Response to comment: Sorry for the misleading expression. This sentence does not mean the scaling factors are driving the conclusions. Even without conducting the scaling procedure, our conclusion still hold qualitatively (Fig. S11).

As a matter of fact, we intended to deliver that the magnitude of observationally-constrained projection of changes in compound hot extremes are pronouncedly greater than that from raw CMIP5 projections. This comparison implies that future risks and impacts related to the worsening of summertime compound hot extremes might be much severer than expected.

This part has been re-written to deliver the point more clearly.

Reviewers' comments:

Reviewer #1 (Remarks to the Author):

Review of revised manuscript "Anthropogenically-driven increases in the risks of summertime compound hot extremes" by Wang et al.

The authors have carefully considered the reviews on their manuscript and made appropriate changes. They have extended their analysis substantially to increase its usefulness to the community.

My only remaining comment is that a little more discussion on the implications of recent work for projections of compound heat extremes could be extended. In the authors' response to reviewer 2 they briefly raise this point but it is not discussed in the revised manuscript as far as I can see. Projections of summertime temperatures in regions of strong land-atmosphere coupling are highly variable between models and strongly related to precipitation change (King, 2019; Vogel et al., 2018). The Figure 6 timeseries suggest increasing spread in projections of compound heat extremes with greater spread than in individual hot days and nights. The increasing difference in projections of compound heat extremes between models may be linked to diverging projections of precipitation. I think this would be worthy of brief discussion and would likely motivate further studies following the results of this study and other recent work.

References

King, A. D. (2019). The drivers of nonlinear local temperature change under global warming. *Environmental Research Letters*. <https://doi.org/10.1088/1748-9326/ab1976>

Vogel, M. M., Zscheischler, J., & Seneviratne, S. I. (2018). Varying soil moisture–atmosphere feedbacks explain divergent temperature extremes and precipitation projections in central Europe. *Earth System Dynamics*, 9(3), 1107–1125. <https://doi.org/10.5194/esd-9-1107-2018>

Reviewer #2 (Remarks to the Author):

The authors have thoroughly responded to the first round of reviews and added some interesting analysis that I think will make the paper suitable for publication after some minor changes.

In particular, the authors show that the increase in compound extremes seems to be linked both to an increase in circulation patterns conducive to hot daytime temperatures as well as to anticorrelations between T_{min} and precipitation. This seems to suggest different processes are acting on T_{max} and T_{min}, and in some places where both T_{max} and T_{min} rise, compound extremes increase in frequency more than either T_{max} or T_{min} alone. That seems novel and interesting to me.

I think the remaining piece here is the link to human impacts - the authors discuss trying unsuccessfully to get mortality datasets, which I know are hard to come by. Short of that, though, I think linking these to population projections would give the reader some sense of how many people will be affected by compound extremes.

There are spatially explicit population projections as a part of the Shared Socioeconomic Pathways project which the authors could use (see: <https://iopscience.iop.org/article/10.1088/1748-9326/11/8/084003>). I think a small analysis showing the historical and future growth of population exposure to compound extremes, T_{max} extremes, and T_{min} extremes would be interesting and would highlight the potential impacts of these. With this analysis, I think the paper would tell a complete story: 1) compound extremes are observed to have increased; 2) a mechanism for this increase is proposed; 3) the projected human impacts of future compound extremes are assessed.

I think after those changes this would be a compelling paper which should be published in NComms.

Reviewer #3 (Remarks to the Author):

thank you for addressing my concerns

General replies to Editor's and Reviewers' comments

We are happy to know that our responses and revisions are satisfactory to Editor and reviewers. Again, we sincerely appreciate your highly constructive comments in the second round of review.

Following Reviewer#1's suggestion, we have now extended discussions about potential sources of uncertainties of compound event projections. Taking Reviewer#2's suggestions into account, we estimate future population exposure to compound hot extremes, which expand the usefulness of our observation-detection-attribution-projection outcomes. We also effectively reduce redundancy and ambiguity throughout the manuscript to keep it as concise and clear as possible. Moreover, several members in the author team double-checked the codes for calculating and plotting to guarantee that we present right results and interpretations. We found a small error in re-gridding and masking the model simulations. Now it has been corrected, with all simulation-relevant figures replaced and some explanations adjusted slightly. Such a minor correction doesn't cause any significant and qualitative changes in our conclusions. We'd happy to share these calculating and plotting codes if required.

We also re-structure the manuscript a little bit to comply with the format requirements of *Nature Communication*. Specifically, we move the brief summary, which was in the first paragraph of the Discussion section, to the last paragraph of the Introduction Section, as required.

Point-by-point responses are presented as follows with reviewers' comments in black and our responses in blue. In the revised manuscript, all major revisions are highlighted in blue as well.

Reviewer #1:

My only remaining comment is that a little more discussion on the implications of recent work for projections of compound heat extremes could be extended. In the authors' response to reviewer 2 they briefly raise this point but it is not discussed in the revised manuscript as far as I can see. Projections of summertime temperatures in regions of strong land-atmosphere coupling are highly variable between models and strongly related to precipitation change (King, 2019; Vogel et al., 2018). The Figure 6 timeseries suggest increasing spread in projections of compound heat extremes with greater spread than in individual hot days and nights. The increasing difference in projections of compound heat extremes between models may be linked to diverging projections of precipitation. I think this would be worthy of brief discussion and would likely motivate further studies following the results of this study and other recent work.

Response to comment: We appreciate this kind suggestion. Indeed, both our mechanism explanations (Fig. 3, Supplementary Fig. 6 and Supplementary Fig. 7) and other recent studies imply that the increasing large spread of compound event projections may be linked to diverging projections of precipitation and resultant discrepancies in land-air interaction physics. Following your suggestions, we have added brief discussions and some outlooks in this regard at the end of the Discussion Section, with the recommended two papers cited.

Reviewer #2:

The authors have thoroughly responded to the first round of reviews and added some interesting analysis that I think will make the paper suitable for publication after some minor changes. In particular, the authors show that the increase in compound extremes seems to be linked both to an increase in circulation patterns conducive to hot daytime temperatures as well as to anticorrelations between T_{min} and precipitation. This seems to suggest different processes are acting on T_{max} and T_{min} , and in some places where both T_{max} and T_{min} rise, compound extremes increase in frequency more than either T_{max} or T_{min} alone. That seems novel and interesting to me.

Response to comment: Thank you for your encouraging comments. Your suggestions in the previous round of review really enlightened us a lot.

I think the remaining piece here is the link to human impacts - the authors discuss trying unsuccessfully to get mortality datasets, which I know are hard to come by. Short of that, though, I think linking these to population projections would give the reader some sense of how many people will be affected by compound extremes. There are spatially explicit population projections as a part of the Shared Socioeconomic Pathways project which the authors could use (see: <https://iopscience.iop.org/article/10.1088/1748-9326/11/8/084003>). I think a small analysis showing the historical and future growth of population exposure to compound extremes, T_{max} extremes, and T_{min} extremes would be interesting and would highlight the potential impacts of these. With this analysis, I think the paper would tell a complete story: 1) compound extremes are observed to have increased; 2) a mechanism for this increase is proposed; 3) the projected human impacts of future compound extremes are assessed. I think after those changes this would be a compelling paper which should be published in NComms.

Response to comment: Thank you very much for this kind suggestion and the useful information about population projection data. Indeed, assessing future population exposure is a good idea in giving readers general sense of implications of future changes in heat hazard on human society. Adding this part of analysis also make our paper more complete logically.

Changes made in response to comment: Following your suggestion, we have added a text section, a main figure (Fig. 7) and a supplementary figure (Supplementary Fig. 13) to analyze historical and future changes in population exposure to summertime hot extremes. The description of population projection data and its pre-processing as well as other technical details have been added in corresponding sections in the revised version.

Following the study of Jones et al. (2015), we measure population exposure by the number of person-days experiencing hot extremes, which accounts for both population dynamics and hazard increases⁴² in the changing population exposure. This index is calculated as the summer number of events multiplied by the number of people exposed. The calculation is undertaken at grid-box scale in individual simulation, hemisphere-scale average for that simulation, multi-member mean for each participating model and multi-model ensemble (MME) mean in sequence. This procedure mandates the use of raw climate projections which provide outputs available at a grid-box scale. Following the study of Vuuren and Carter (2014), we select a RCP4.5-SSP1 combination to frame a world evolving into a future with relatively low challenges to adaptation and mitigation, and a RCP8.5-SSP3 combination to characterize a world with rapid growth in emissions and population, i.e. the most challenging scenario.

The results indicate that even if the world evolves toward a generally sustainable future (RCP4.5-SSP1), the Northern Hemisphere still expects to see nearly a quadrupling of population exposure to compound hot extremes (19.5 billion person-days to 74.0 billion person-days) (Fig. 7a). By contrast, the most challenging scenario (RCP8.5-SSP3) is projected to see an over eightfold increase to 172.2 billion person-days by then end of the 21st century (Fig. 7b), with hotspots clustered over populous regions such

as eastern United States, western Europe, western Asia and eastern China (Supplementary Fig. 13). Population exposure to daytime and nighttime hot extremes exhibits a similar peak structure, with the difference between two worlds substantially smaller than the compound type (Fig. 7 and Supplementary Fig. 13).

In the revised manuscript, we clearly state that these results only represent a lower boundary of exact estimate for future population exposure to heat hazards, since the used raw climate projections underestimate future increases in compound heat hazards as elaborated in the “Observationally-constrained projection” section. Meanwhile, to keep consistent with other parts of analysis through the study, we also mask the population projection by the availability of observational dataset. So, underestimation in population exposure to compound hot extremes also arises from the insufficient land coverage in the analysis, with some highly populous areas like India unaccounted for (Supplementary Fig. 13).

Also following your suggestion, we introduce the storyline of our analysis more clearly before presenting the main results in the revised version, i.e. following observational facts—mechanisms & drivers—projection of hazards and human exposure.

Newly added references:

42. Jones, B., O'Neill, B. C., McDaniel, L., McGinnis, S., Mearns, L. O. & Tebaldi, C. Future population exposure to US heat extremes. *Nature Clim. Change* **5**, 652–655 (2015).
43. Jones, B. & O'Neill, B. C. Spatially explicit global population scenarios consistent with the Shared Socioeconomic Pathways. *Environ. Res. Lett.* **11**, 084003 (2016).
65. van Vuuren, D. P. & Carter, T. R. Climate and socio-economic scenarios for climate change research and assessment: reconciling the new with the old. *Clim. Change* **122**, 415–429 (2014).

Fig. 7 Projection of population exposure to summertime hot extremes across the Northern Hemisphere land areas. **a** Population exposure to summertime compound hot extremes (purple lines), independent hot days (blue lines), and independent hot nights (green lines) in the twenty-first century in the integrated scenario combining RCP4.5 (climate) and SSP1 (population). **b** Same as **a**, but in integrated scenario constituted by RCP8.5 (climate) and SSP3 (population). Decadal-average multi-model ensemble (MME) mean are indicated by solid dots connected by solid curves, with vertical bars framing the 5-95% range of all members' projections of population exposure. The vertical dashed line locates the year of 2030, after which compound hot extremes will become the type that populations in the Northern Hemisphere are most frequently exposed to.

Supplementary Fig. 13 Projected changes in population exposure to summertime hot extremes between the decade of 2010s and 2090s (the 2090s minus the 2010s). **a, b** Compound hot extremes; **c, d** Independent hot days; **e, f** Independent hot nights. Shown are MME mean projected changes in two integrated scenarios designed as RCP4.5 (climate)-SSP1 (population) combination in **a-c**, and RCP8.5 (climate)-SSP3 (population) combination in **d-f**.

Reviewer #3:

Thank you for addressing my concerns

Response to comment: Thank you very much for your time and efforts devoted to help improving this paper.

REVIEWERS' COMMENTS:

Reviewer #2 (Remarks to the Author):

My comments from the last round have been addressed by the authors and I think the paper is ready for publication.